# PROTOREG: PRIORITIZING DISCRIMINATIVE INFORMATION FOR FINE-GRAINED TRANSFER LEARNING

## ABSTRACT

Transfer learning leverages a pre-trained model with rich features to fine-tune it for downstream tasks, thereby improving generalization performance. However, we point out the "*granularity gap*" in fine-grained transfer learning, a mismatch between the level of information learned by a pre-trained model and the semantic details required for a fine-grained downstream task. Under these circumstances, excessive non-discriminative information can hinder the sufficient learning of discriminative semantic details. In this study, we address this issue by establishing class-discriminative prototypes and refining the prototypes to gradually encapsulate more fine-grained semantic details, while explicitly aggregating each feature with the corresponding prototype. This approach allows the model to prioritize fine-grained discriminative information, even when the pre-trained model contains excessive non-discriminative information due to the granularity gap. Our proposed simple yet effective method, ProtoReg, significantly outperforms other transfer learning methods in fine-grained classification benchmarks with an average performance improvement of 6.4% compared to standard fine-tuning. Particularly in limited data scenarios using only 15% of the training data, ProtoReg achieves an even more substantial average improvement of 13.4%. Furthermore, ProtoReg demonstrates robustness to shortcut learning when evaluated on out-of-distribution data.

## 1 INTRODUCTION

Training a model from scratch suffers from overfitting to non-discriminative signals since the model has no prior knowledge of task-specific discriminative information (Goodfellow et al., 2016; Geirhos et al., 2020). The model can be biased towards easy-to-fit and non-discriminative patterns, such as high-frequency patterns or shortcuts, that disrupt generalized learning for the task (Ilyas et al., 2019; Xiao et al., 2020; Geirhos et al., 2018; Izmailov et al., 2022). Transfer learning can alleviate such issues by leveraging information learned by a pre-trained model. Consequently, transfer learning enables a model to achieve better generalization performance compared to training the model from scratch (Kornblith et al., 2019b; Donahue et al., 2014).

We highlight a potential challenge in transfer learning when it is applied to a downstream task with finer granularity, demanding detailed discriminative information, which leads to the emergence of a granularity gap compared to the pre-trained knowledge. Due to the granularity gap, the coarse information learned by pre-trained models often does not align with the discriminative signals required to solve fine-grained downstream tasks. For example, when transferring to classify aircraft types (FGVC Aircraft (Maji et al., 2013)), an ImageNet pre-trained model may capture general information about the overall appearance of aircraft but may lack class-discriminative attributes of a particular aircraft model, such as swept-back wings and vertical tail fins, which are characteristic of the Boeing 707-200 model. The presence of excessive non-discriminative information can lead the model to overfit non-discriminative information in the training data, hindering effective fine-grained transfer learning. Figure 1 illustrates that the retrieved images using features from the pre-trained model belong to different classes than the query image, while sharing coarse and non-discriminative information (such as color or background). In such scenarios, fine-tuning with cross-entropy (CE) loss alone fails to adequately capture class-discriminative information (e.g., the distinct facial patterns of a cardinal) without any constraints to prioritize class-discriminative information.

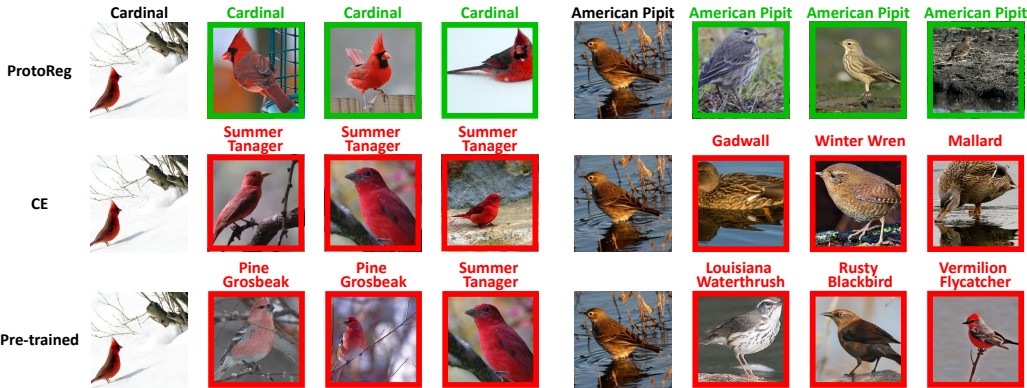

Figure 1: Image retrieval results on CUB-200-2011 using a ResNet-50 pre-trained on ImageNet. The three cases include: direct use of the pre-trained model (bottom), fine-tuning with cross-entropy loss only (middle), and applying the proposed ProtoReg with cross-entropy loss (top). In the pre-trained representation space, the images retrieved from the query image share coarse and non-discriminative information (left: red color, right: water background). Although fine-tuning using only cross-entropy loss faces challenges in capturing fine-grained class-discriminative information, our proposed regularizer alleviates this issue.

In this study, we introduce a simple yet effective method that aims to enhance transferability to downstream tasks by prioritizing fine-grained and class-discriminative information when excessive non-discriminative information exists in the pre-trained model due to the granularity gap. To this end, we introduce adaptively evolving prototypes for each class, which are initialized to contain class-discriminative information. During transfer learning, we regularize the feature space to align the features with their corresponding prototypes, encouraging the features to contain more class-discriminative information. Furthermore, as the model learns additional class-discriminative information required for the downstream task, we enhance the prototypes to adaptively incorporate the class-discriminative information.

Our proposed method, ProtoReg, significantly improves performance across various fine-grained visual classification tasks, especially outperforming in limited data scenarios that are particularly vulnerable to overfitting of non-discriminative information. For instance, it achieves an average performance improvement of 6.4% on four fine-grained downstream datasets and an average improvement of 13.4% when using only 15% of the training data from the same datasets. Additionally, to verify whether the model performs robust predictions based on the class-discriminative information of objects, we evaluate the model fine-tuned on the CUB-200-2011 (Wah et al., 2011) using the Waterbirds (Sagawa et al., 2019) test dataset. This dataset intentionally changes the background in which the birds are placed to disrupt correlations between the background and the objects. Our proposed method exhibited a notably smaller accuracy drop when compared to other methods, underscoring the model's efficacy in acquiring fine-grained class-discriminative information.

## 2 RELATED WORK

### 2.1 TRANSFER LEARNING

Two standard approaches for transfer learning are linear probing and fine-tuning. Linear probing involves freezing the encoder responsible for feature extraction and training only a new linear classifier for the downstream task. On the other hand, fine-tuning adjusts the entire model for the downstream task, typically outperforming linear probing (Chen et al., 2021; Yosinski et al., 2014). A recent study (Kumar et al., 2022) has pointed out that fine-tuning can lead to feature distortion in pre-trained models, and it proposes LP-FT as a method that initially employs linear probing and subsequently fine-tunes the entire model.

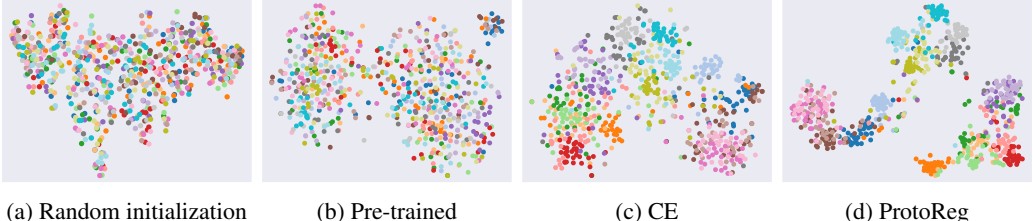

|   (a) Random initialization   |   (b) Pre-trained   |   (c) CE   |   (d) ProtoReg   |

Figure 2: T-SNE visualization of validation features for FGVC Aircraft using (a) a randomly initialized model, (b) a pre-trained model, (c) a model fine-tuned with cross-entropy loss, and (d) a model fine-tuned jointly with ProtoReg. All models are ResNet-50, with the pre-trained model trained on ImageNet-1K. We visualized the first 20 classes, where each color represents a specific class.

## 2.2 REGULARIZATION METHODS FOR TRANSFER LEARNING

To prevent the over-fitting in transfer learning, various regularization methods have been developed. L2-SP (Xuhong et al., 2018) constrains the fine-tuned model weights to be close to their corresponding pre-trained weights. DELTA (Li et al., 2019) regularizes the behavior of the fine-tuned model by decreasing the feature map distances between the pre-trained and fine-tuned models. To alleviate negative transfer, BSS (Chen et al., 2019) penalizes the small singular values of the penultimate layer features. On the other hand, SN (Kou et al., 2020) replaces Batch Normalization (Ioffe & Szegedy, 2015) with its stochastic variant to transfer the running statistics. By additionally tuning a pre-trained linear classifier leveraging the relationship between the source and target classes, Co-tuning (You et al., 2020) aims to fully fine-tune the entire model to improve transferability. Although various regularization methods have been proposed for transfer learning, direct consideration of the granularity gap has been lacking. Our approach introduces a regularization method aimed at prioritizing fine-grained discriminative signals, particularly in scenarios where an abundance of non-class-discriminative information exists in the pre-trained information due to the granularity gap.

## 3 MOTIVATIONAL OBSERVATIONS

Figure 2a and 2b illustrate the t-SNE visualization (Van der Maaten & Hinton, 2008) of the FGVC Aircraft (Maji et al., 2013) validation features obtained from the random initialized and pre-trained ResNet-50 (He et al., 2016) models, respectively. The pre-trained model was trained on the ImageNet-1K dataset (Deng et al., 2009) and we used penultimate features to visualize. Each color in the figure corresponds to a different class. In contrast to the previous findings that the feature representations of downstream data with similar granularity, such as CIFAR10 (Krizhevsky et al., 2009), acquired from the ImageNet pre-trained model exhibit a high degree of linear separability (Chen et al., 2020; Kumar et al., 2022), Figure 2b demonstrates that the pre-trained model remains susceptible to non-class-discriminative features in the context of the fine-grained classification.

Furthermore, Figure 2a and 2b show a comparable degree of class separation, indicating overfitting to non-class-discriminative signals, commonly encountered when training a randomly initialized model, can also arise when transferring pre-trained knowledge to a fine-grained downstream task. Figure 2c displays the visualization of the validation features after fine-tuning using only cross-entropy loss. Despite achieving a training accuracy of 100%, the clusters of validation features exhibit noisy separation between classes, signifying that the model acquired limited class-discriminative features. This implies that the model has overfit to non-class-discriminative patterns, rather than capturing the intricate semantic patterns that distinguish between fine-grained classes during fine-tuning.

These observations underscore the importance of pre-trained models prioritizing their class-discriminative information during fine-tuning, especially when addressing fine-grained downstream tasks that exhibit a granularity gap with the pre-trained knowledge. In the following section, we will describe our approach, ProtoReg, designed to effectively regulate the model to facilitate improved acquisition of class-discriminative information during transfer learning. ProtoReg enhances learning of class-discriminative information, leading to improved discriminability of validation features (Figure 2d).

# 4 METHOD

## 4.1 PROBLEM FORMULATION

We consider a classification task as the downstream task for transfer learning. The pre-trained model is composed of a feature extractor $\mathcal{F}_{\boldsymbol{\theta}}$ and, in cases where the pre-trained model has been trained through classification, a linear classifier $\mathcal{H}_{\boldsymbol{\psi}}$. During the transfer for the downstream task, $\mathcal{H}_{\boldsymbol{\psi}}$ is removed. Subsequently, a new linear classifier $\mathcal{G}_{\boldsymbol{\phi}}$ is introduced for downstream task and the entire model $\mathcal{G}_{\boldsymbol{\phi}} \circ \mathcal{F}_{\boldsymbol{\theta}}$ is fine-tuned. The downstream data $\mathcal{D}$ consists of samples belonging to $K$ classes. $\mathcal{D}_i$ denotes a set of samples from class $i$, and $\boldsymbol{x}_{i,n}$ denotes the $n$-th sample from class $i$:

$$\mathcal{D} = \bigcup_{i=1}^{K} \mathcal{D}_i, \quad \mathcal{D}_i = \bigcup_{n=1}^{N_i} \{(\boldsymbol{x}_{i,n}, i)\}, \tag{1}$$

where $N_i$ is the number of samples in class $i$. $\mathcal{D}$ is split into a disjoint set $\mathcal{D}^{\text{train}}$, $\mathcal{D}^{\text{val}}$, and $\mathcal{D}^{\text{test}}$ for training, validation, and testing.

## 4.2 PROTOTYPE INITIALIZATION

To prioritize and focus on the class-discriminative information during fine-tuning, we first initialize class-discriminative prototypes $\{\boldsymbol{c}_i\}_{i=1}^{K}$, where $\boldsymbol{c}_i$ denotes the prototype of class $i$. An intuitive approach for defining prototypes is to utilize the sample mean of features for each class (Snell et al., 2017) such that

$$\boldsymbol{c}_i = \frac{1}{N_i^{\text{train}}} \sum_{n=1}^{N_i^{\text{train}}} \boldsymbol{z}_{i,n} \in \mathbb{R}^d, \tag{2}$$

where $\boldsymbol{z}_{i,n}$ denotes the encoded feature of $\boldsymbol{x}_{i,n}$ (i.e., $\boldsymbol{z}_{i,n} := \mathcal{F}_{\boldsymbol{\theta}}(\boldsymbol{x}_{i,n})$), $N_i^{\text{train}}$ denotes the number of training samples of class $i$, and $d$ denotes the feature dimension. The linear classifier $\mathcal{G}_{\boldsymbol{\phi}}$ is randomly initialized. However, when samples from different classes share non-discriminative information, as shown in Figure 1, using the sample mean may not adequately encompass class-discriminative information. To enhance the incorporation of class-discriminative information into the prototypes, we introduce an alternative method for their initialization. This involves utilizing the discriminative classifier weights obtained from linear probing, as follows:

$$\hat{\boldsymbol{\phi}} = \arg\max_{\boldsymbol{\phi}} \text{ACC}(\mathcal{D}^{\text{val}}; \mathcal{G}_{\boldsymbol{\phi}} \circ \mathcal{F}_{\bar{\boldsymbol{\theta}}}), \quad \boldsymbol{c}_i = \hat{\boldsymbol{\phi}}_{i,:}, \tag{3}$$

where ACC denotes the classification accuracy, $\bar{\boldsymbol{\theta}}$ denotes the frozen encoder parameters, and $\hat{\boldsymbol{\phi}}_{i,:}$ denotes the $i$-th row of $\hat{\boldsymbol{\phi}} \in \mathbb{R}^{K \times d}$. To determine $\boldsymbol{\phi}$ that maximizes validation accuracy, we use early stopping when the validation accuracy begins to decrease. The prototypes initialized from the linear classifier weights, serving as decision boundaries that separate downstream classes in a pre-trained feature space, explicitly provide downstream task-specific discriminative information contained in the pre-trained model to the downstream model during fine-tuning. To utilize the linear classifier $\mathcal{G}_{\boldsymbol{\phi}}$ as the class prototypes directly, we trained the classifier without using a bias term.

## 4.3 PROTOTYPE REFINEMENT

During fine-tuning, the model is trained to progressively encode task-discriminative information. To ensure that the class-discriminative prototypes adapt to the additional information learned, we refine prototypes during fine-tuning. When the prototypes are initialized based on class-wise mean features (Eq 2), we utilize a similar approach for prototype refinement. Specifically, we introduce class-wise memory banks $\{\mathcal{M}_i^t\}_{i=1}^{K}$ to refine prototypes at the end of each epoch. For every iteration at epoch $t$, the batch features are enqueued to their corresponding class-wise memory banks. At the end of epoch $t$, the updated prototype $\boldsymbol{c}_i^{t+1}$ is determined as the sample mean of the enqueued features in $\mathcal{M}_i^t$ as follows:

$$\boldsymbol{c}_i^{t+1} = \frac{1}{N_i^{\text{train}}} \sum_{n=1}^{N_i^{\text{train}}} \mathcal{M}_i^t(n) \in \mathbb{R}^d, \tag{4}$$

where $\mathcal{M}_i^t(n)$ denotes the $n$-th feature enqueued in $\mathcal{M}_i^t$. After the refinement, the memory banks are flushed and updated as $\{\mathcal{M}_i^{t+1}\}_{i=1}^K$ where the features at epoch $t+1$ are enqueued iteratively. Refining prototypes by averaging over samples across progressively shifted features spaces is beneficial for stable training, given the gradual shift towards task-discriminative feature spaces.

When initializing prototypes based on linear classifier weights (Eq 3), we do not utilize updated features to refine prototypes. Instead, we set the prototypes to be learnable and refine them through backpropagation while optimizing Eq 7. Both refinement methods play a crucial role in encouraging prototypes to adaptively evolve to incorporate class-discriminative information, ensuring stable class-discriminative feature learning during the fine-tuning phase.

### 4.4 REGULARIZATION BY PROTOTYPE AGGREGATION AND SEPARATION

To prioritize fine-grained discriminative information when the pre-trained model contains excessive non-class-discriminative information, we explicitly aggregate features towards their corresponding class-discriminative prototype by increasing the cosine similarity as follows:

$$\mathcal{L}_{\text{aggr}}(\boldsymbol{z}_{i,n}) = -\text{sim}(\boldsymbol{z}_{i,n}, \boldsymbol{c}_i), \tag{5}$$

where $\text{sim}(\cdot, \cdot)$ is a cosine similarity function. The aggregation term encourages features to be more focused on class-discriminative information. Additionally, to mitigate reliance on class-shared information, such as red color which is non-discriminative information across some bird species in Figure 1, we employ a separation loss that enforces features to be dissimilar from prototypes of different classes as follows:

$$\mathcal{L}_{\text{sep}}(\boldsymbol{z}_{i,n}) = \log \sum_{s \neq i} \exp(\text{sim}(\boldsymbol{z}_{i,n}, \boldsymbol{c}_s)). \tag{6}$$

Consequently, $\mathcal{L}_{\text{aggr}}$ and $\mathcal{L}_{\text{sep}}$ are jointly optimized with the cross-entropy loss $\mathcal{L}_{\text{ce}}$ to encourage the model to explicitly prioritize class-discriminative information during fine-tuning:

$$\mathcal{L}_{\text{total}} = \mathcal{L}_{\text{ce}} + \lambda_{\text{aggr}}\mathcal{L}_{\text{aggr}} + \lambda_{\text{sep}}\mathcal{L}_{\text{sep}}, \tag{7}$$

where $\lambda_{\text{aggr}}$ and $\lambda_{\text{sep}}$ are the coefficient hyper-parameters for $\mathcal{L}_{\text{aggr}}$ and $\mathcal{L}_{\text{sep}}$, respectively. We provide the pseudo code for ProtoReg in Appendix A.1.

## 5 EXPERIMENTS

### 5.1 DATASETS AND IMPLEMENTATION DETAILS

We conducted experiments on six well-known image classification benchmarks, which include four fine-grained datasets (FGVC Aircraft (Maji et al., 2013), Stanford Cars (Krause et al., 2013), CUB-200-2011 (Wah et al., 2011), NABirds (Van Horn et al., 2015)) and two general datasets (Caltech101 (Li et al., 2022), CIFAR100 (Krizhevsky et al., 2009)). In order to simulate scenarios where models are more susceptible to overfitting to non-class-discriminative information, we randomly sampled each training dataset with four different sampling rates of 15%, 30%, 50%, and 100%, adopting evaluation protocol of prior studies (Kou et al., 2020; You et al., 2020).

In all experiments, we used ResNet-50 (He et al., 2016) pre-trained on ImageNet-1K (Deng et al., 2009), and the pre-trained weights were obtained from Torchvision (Falbel, 2022). We report the test accuracy using the model with the highest accuracy on the validation data. In case the validation set is not available in the benchmark, we randomly selected 20% of the training data to construct the validation data, which was performed before sampling the train data. More detailed explanations regarding datasets and implementation details can be found in Appendix A.2.

### 5.2 MAIN RESULTS

Table 1 presents the overall performance comparison of ProtoReg with the compared methods. We refer to the case where we initialize prototypes using class-wise mean features (Eq 2) as ProtoReg (self), and the case using linear classifier weights (Eq 3) as ProtoReg (LP). Our proposed method outperforms all the compared methods by a large margin across all benchmarks and sampling rates.

Table 1: The overall performance of the test accuracy (%) on the four benchmarks. The network is ResNet-50 pre-trained on ImageNet-1K. The baseline is fine-tuning with cross-entropy loss (CE). The compared methods are L2-SP (Xuhong et al., 2018), BSS (Chen et al., 2019), SN (Kou et al., 2020), Co-tuning (You et al., 2020), LP-FT (Kumar et al., 2022), Robust FT (Xiao et al., 2023), and DR-Tune (Zhou et al., 2023).

| Dataset | Method | Sampling rate | | | |
|---------|--------|-----|-----|-----|------|
| | | 15% | 30% | 50% | 100% |
| FGVC Aircraft | CE | 23.91 | 38.14 | 49.54 | 66.01 |
| | L2-SP | 24.00 | 37.87 | 49.29 | 65.39 |
| | BSS | 24.31 | 38.75 | 50.34 | 66.72 |
| | SN | 26.49 | 41.56 | 52.39 | 68.71 |
| | Co-tuning | 26.60 | 40.89 | 52.57 | 68.67 |
| | LP-FT | 24.39 | 38.19 | 49.36 | 66.76 |
| | Robust FT | 23.91 | 37.84 | 49.09 | 65.08 |
| | DR-Tune | 24.42 | 40.89 | 50.65 | 68.80 |
| | ProtoReg (self) | 33.66 | 46.83 | 59.95 | 75.25 |
| | ProtoReg (LP) | **34.35** | **50.41** | **61.45** | **77.89** |
| Stanford Cars | CE | 25.75 | 52.66 | 70.26 | 84.54 |
| | L2-SP | 25.81 | 52.63 | 70.21 | 84.31 |
| | BSS | 26.28 | 54.06 | 71.54 | 85.10 |
| | SN | 29.62 | 56.88 | 72.74 | 84.54 |
| | Co-tuning | 29.88 | 57.46 | 74.38 | 86.10 |
| | LP-FT | 27.50 | 53.07 | 70.25 | 84.47 |
| | Robust FT | 25.07 | 52.63 | 70.10 | 84.29 |
| | DR-Tune | 26.80 | 56.04 | 73.55 | 84.80 |
| | ProtoReg (self) | 39.95 | 65.91 | 78.36 | 87.74 |
| | ProtoReg (LP) | **42.52** | **69.32** | **81.62** | **89.60** |
| CUB-200-2011 | CE | 45.16 | 57.94 | 69.56 | 78.12 |
| | L2-SP | 45.19 | 57.74 | 69.61 | 78.24 |
| | BSS | 45.89 | 59.77 | 70.62 | 79.24 |
| | SN | 49.89 | 62.48 | 71.11 | 79.28 |
| | Co-tuning | 50.35 | 63.43 | 71.12 | 79.99 |
| | LP-FT | 48.48 | 59.91 | 69.45 | 78.69 |
| | Robust FT | 43.34 | 57.66 | 68.35 | 78.53 |
| | DR-Tune | 44.84 | 58.49 | 69.62 | 79.10 |
| | ProtoReg (self) | 57.09 | 67.48 | 74.53 | 81.22 |
| | ProtoReg (LP) | **59.08** | **69.49** | **76.30** | **82.38** |
| NABirds | CE | 39.87 | 55.80 | 65.89 | 74.98 |
| | L2-SP | 39.63 | 55.68 | 65.32 | 74.95 |
| | BSS | 40.53 | 57.07 | 66.43 | 75.96 |
| | SN | 43.86 | 57.95 | 66.72 | 75.44 |
| | Co-tuning | 44.85 | 60.85 | 69.20 | 77.41 |
| | LP-FT | 41.06 | 56.09 | 66.00 | 74.79 |
| | Robust FT | 39.00 | 55.17 | 65.46 | 74.86 |
| | DR-Tune | 41.37 | 57.07 | 66.99 | 76.16 |
| | ProtoReg (self) | 50.75 | 63.68 | 70.84 | 78.40 |
| | ProtoReg (LP) | **52.42** | **64.83** | **72.33** | **79.51** |

For example, ProtoReg (LP) achieves a test accuracy of 77.89% on the FGVC Aircraft at a 100% sampling rate, outperforming the baseline by 11.88% and the best-performing compared method by 9.18%. Notably, in limited data scenarios where models are more susceptible to overfitting to non-class-discriminative information, ProtoReg shows significant improvements compared to the other methods. For example, when using only 15% of the training samples of the Stanford Cars, ProtoReg outperforms the best-performing compared method by 6.74%p with ProtoReg (self) and 8.73%p with ProtoReg (LP). On average, it exhibits a 6.4% performance improvement compared to standard fine-tuning across four fine-grained downstream tasks and achieves an average performance improvement of 13.4% in limited data scenarios using only 15% of the training data on each dataset.

## 5.3 EFFECTIVE LEARNING OF CLASS-DISCRIMINATIVE INFORMATION

To verify that ProtoReg effectively learns class-discriminative information, we evaluated a model trained on CUB-200-2011 on the Waterbirds (Sagawa et al., 2019) test dataset. The dataset consists of bird images from the CUB-200-2011 dataset with backgrounds replaced by other scenes from the Places (Zhou et al., 2017). This modification intentionally disrupts the correlation between objects

Table 2: Test accuracy (%) on CUB-200-2011 (ID) and Waterbirds (OOD) test datasets for a model trained on CUB-200-2011. ↓ represents the drop rate (%) in OOD test accuracy compared to ID test accuracy.

| Method | Sampling rates | | | | | | | | | | | |
| | 15% | | | 30% | | | 50% | | | 100% | | |
| | ID | OOD | ↓ | ID | OOD | ↓ | ID | OOD | ↓ | ID | OOD | ↓ |
|---|---|---|---|---|---|---|---|---|---|---|---|---|
| CE | 45.2 | 23.1 | 49 | 57.9 | 36.5 | 37 | 69.6 | 47.5 | 32 | 78.1 | 60.2 | 24 |
| Co-tuning | 50.4 | 34.3 | 32 | 63.4 | 48.0 | 24 | 71.1 | 58.7 | 17 | 80.0 | 68.9 | 14 |
| LP-FT | 48.5 | 26.3 | 46 | 59.9 | 36.0 | 40 | 69.5 | 47.7 | 31 | 78.7 | 58.8 | 25 |
| ProtoReg (self) | 57.1 | 42.8 | 25 | 67.5 | 52.8 | 22 | 74.5 | 59.6 | 20 | 81.2 | 70.3 | 13 |
| ProtoReg (LP) | 59.0 | **47.5** | **19** | 69.5 | **57.8** | **17** | 76.3 | **65.2** | **13** | 82.4 | **74.3** | **10** |

Figure 3: Predicted class probabilities for a sample from CUB-200-2011 (left) and its corresponding sample from Waterbirds (right). The ground truth is American Redstart. For the left in-distribution sample, both CE and ProtoReg accurately predict the ground truth. However, when the background is changed while maintaining the object's fine-grained class-discriminative information, ProtoReg still predicts accurately, while CE makes overconfident incorrect predictions.

and backgrounds in bird images. As a result, models that fail to adequately learn fine-grained class-discriminative information and rely more on coarse information are expected to encounter a more significant accuracy drop on the out-of-distribution (OOD) dataset.

Table 2 provides a comparison of test accuracy for CUB-200-2011 (ID) and Waterbirds (OOD) across four different sampling rates. The model fine-tuned with ProtoReg exhibits better robustness to changes in the background, with an accuracy drop rate of only 10% at a sampling rate of 100%, compared to the drop rate of 24% with cross-entropy loss alone. Especially in a limited data scenario where only 15% of the training data is used, our method demonstrates an accuracy drop of only 19%, surpassing both the baseline with a 49% drop and the best-compared method with a 32% drop in OOD test accuracy.

Figure 3 illustrates the predicted probabilities for a sample from CUB-200-2011 and its corresponding sample from the Waterbirds (ground truth: American Redstart). The model fine-tuned using cross-entropy loss shows a significant decrease in the ground truth probability from 0.765 to 0.027. On the other hand, the model regularized with ProtoReg not only accurately predicts the class but also maintains the top-3 predicted class rank. GradCAM (Selvaraju et al., 2017) visualization in Figure 10 demonstrates that ProtoReg effectively captures the discriminative features of the object for prediction, while CE shows a noticeable bias towards the background.

### 5.4 ANALYSIS OF PROTOREG

In this subsection, we analyze the impact of ProtoReg on transfer learning and the effectiveness of each of its components. Unless stated otherwise, the experiments are based on ProtoReg (self) on FGVC Aircraft with a 100% sampling rate. Additional results can be found in Appendix B.

#### 5.4.1 IMPACT ON THE REPRESENTATION SPACE

To show how CE, ProtoReg (self), and ProtoReg (LP) have transformed the representation space of the pre-trained model, Figure 4 presents CKA similarities (Cortes et al., 2012) between the pre-trained model and each fine-tuned model. The CKA similarity measures the similarity between the layer representations of two models. A lower CKA score indicates that the two layers encode differ-

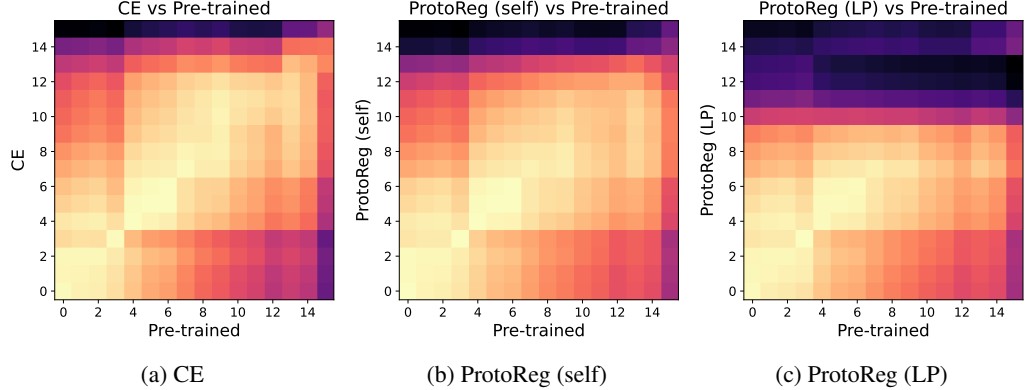

(a) CE        (b) ProtoReg (self)        (c) ProtoReg (LP)

Figure 4: CKA similarities between the pre-trained model and CE, ProtoReg (self), and ProtoReg (LP). The indices along the two axes represent the 16 blocks that comprise ResNet-50. A lower CKA similarity indicates that the two layers encode different information.

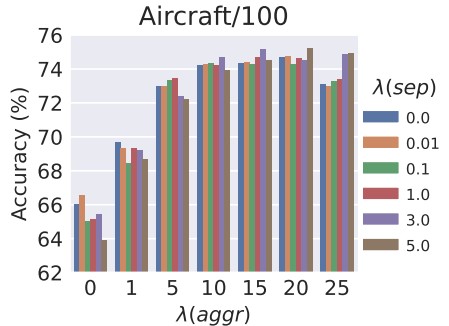

Figure 5: Test accuracy (%) for various combinations of $\lambda_{\mathrm{aggr}}$ and $\lambda_{\mathrm{sep}}$.

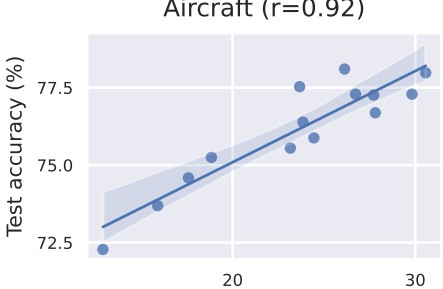

Figure 6: Effects initializing discriminative prototypes for transfer accuracy.

ent information from each other (Kornblith et al., 2019a; Raghu et al., 2021). For CE, the decrease in CKA similarity with the pre-trained model is most pronounced in the penultimate layer, which deals with high-level semantic information. This observation suggests that the model failed to learn substantial new information but rather leaned heavily on the pre-existing pre-trained knowledge. On the other hand, both ProtoReg (self) and ProtoReg (LP) exhibit substantial changes in representation space even in intermediate layers preceding the penultimate layer. This suggests that ProtoReg not only relies on pre-trained knowledge but also acquires additional class-discriminative information.

### 5.4.2 IMPORTANCE OF THE AGGREGATION AND SEPARATION STRENGTHS

Figure 5 presents the test accuracies achieved by various combinations of $\lambda_{\mathrm{aggr}}$ and $\lambda_{\mathrm{sep}}$, which correspond to $\mathcal{L}_{\mathrm{aggr}}$ and $\mathcal{L}_{\mathrm{sep}}$, respectively. As $\lambda_{\mathrm{aggr}}$ increases to a certain level, test accuracy consistently improves, resulting in a performance improvement of more than 8%p. This observation suggests that applying strong aggregation towards their class-discriminative prototypes, which encourages the model to focus on class-discriminative information, leads to enhanced transfer performances. Furthermore, a moderate degree of $\lambda_{\mathrm{sep}}$ further enhances the test accuracy when used with a high $\lambda_{\mathrm{aggr}}$. However, using $\lambda_{\mathrm{sep}}$ alone or with a small $\lambda_{\mathrm{aggr}}$ can degrade the test accuracy. The results on different datasets and sampling rates are provided in Appendix B.2.

### 5.4.3 INITIALIZING DISCRIMINATIVE PROTOTYPES IMPROVES TRANSFER LEARNING

Figure 6 illustrates the correlation between the validation accuracy of linear probing accuracy and the downstream test accuracy in ProtoReg (LP). A higher linear probing validation accuracy indicates that the initialized prototypes are more class-discriminative. We vary the linear probing epoch from 1 to 15, setting the prototypes to be less discriminative when they are trained with fewer epochs.

Table 3: Test accuracy (%) when $\mathcal{L}_{\text{aggr}}$ and $\mathcal{L}_{\text{sep}}$ are applied in different stages: early stage (epoch 1 - 30), middle stage (epoch 31 - 60), or late stage (61 - 100).

| | | | $\mathcal{L}_{\text{aggr}}$ | |
|---|---|---|---|---|
| | | Early | Middle | Late |
| | Early | 72.9 | 70.6 | 69.3 |
| $\mathcal{L}_{\text{sep}}$ | Middle | 72.7 | 70.7 | 69.3 |
| | Late | 71.7 | 71.0 | 70.7 |

Table 4: Ablation study on each component of ProtoReg.

| $\mathcal{L}_{\text{ce}}$ | $\mathcal{L}_{\text{aggr}}$ | $\mathcal{L}_{\text{sep}}$ | Refine | Accuracy (%) |
|---|---|---|---|---|
| ✓ | | | | 66.01 |
| ✓ | | ✓ | | 66.34 |
| ✓ | ✓ | | | 70.93 |
| ✓ | ✓ | ✓ | | 71.23 |
| ✓ | | ✓ | ✓ | 63.79 |
| ✓ | ✓ | | ✓ | 74.71 |
| ✓ | ✓ | ✓ | ✓ | 75.25 |

The experimental results reveal a Pearson correlation of 0.92, indicating that initializing prototypes to be discriminative is highly correlated with the downstream test accuracy.

### 5.4.4 REGULARIZATION IN THE EARLY TRANSFER MATTERS

If the pre-trained model does not initially focus well on the limited fine-grained class-discriminative information it contains, the model can be overfitted to excessive non-class discriminative information, potentially leading to insufficient learning of semantic details. Table 3 illustrates the significance of early-stage feature regularization of ProtoReg by selectively applying $\mathcal{L}_{\text{aggr}}$ and $\mathcal{L}_{\text{sep}}$ during the early, middle, or late stages of the training process. Among the nine combinations tested, incorporating both $\mathcal{L}_{\text{aggr}}$ and $\mathcal{L}_{\text{sep}}$ in the early stage yields the highest performance, outperforming the worst case by 3.6%p, which involves applying $\mathcal{L}_{\text{sep}}$ early and $\mathcal{L}_{\text{aggr}}$ late. Notably, the early-stage application of $\mathcal{L}_{\text{aggr}}$ consistently outperforms its late-stage application.

### 5.5 ABLATION STUDY

Table 4 presents the results of the ablation study. When $\mathcal{L}_{\text{sep}}$ is used without $\mathcal{L}_{\text{aggr}}$ and prototype refinement, the accuracy slightly increases compared to the baseline (row 2). When using only $\mathcal{L}_{\text{aggr}}$ without $\mathcal{L}_{\text{sep}}$ and prototype refinement, the features are solely aggregated towards the initial prototypes from the pre-trained model. This leads to an increase of 4.92%p in performance as the pre-trained model is regularized to focus on task-relevant features (row 3). Moreover, when $\mathcal{L}_{\text{aggr}}$ is used with prototype refinement, which enables the prototypes to evolve from coarse to fine-grained, the performance is further improved by 8.70%p (row 6). Furthermore, the addition of $\mathcal{L}_{\text{sep}}$ allows the prototypes to focus more on the discriminative features between classes, leading to a significant increase of 9.24%p (row 7).

The results in Table 5 demonstrate that ProtoReg serves as a powerful regularizer. In CE, increasing the fine-tuning epoch did not lead to any noticeable improvement in the test accuracy, indicating that no additional discrminative information is learned. However, when using ProtoReg, performance is enhanced by 2.25%p when increasing the adaptation epoch from 100 epochs to 400 epochs.

Table 5: Test accuracy (%) according to the number of fine-tuning epochs.

| Method | Epoch | | | |
|---|---|---|---|---|
| | 100 | 200 | 300 | 400 |
| CE | 66.01 | 65.95 | 66.25 | 67.12 |
| ProtoReg | 75.25 | 76.03 | 76.30 | 77.50 |

## 6 CONCLUSION

We introduce ProtoReg, a simple yet effective method for fine-grained transfer leraning that enhances learning class-discriminative information in the presence of granularity gap due to the excessive non-discriminative information in the pre-trained model. We first initialize class-discriminative prototypes by utilizing either the class-wise feature mean or linear probing classifier. Then the aggregation and separation loss are utilized to prioritize class-discriminative information. During fine-tuning, prototypes are refined to contain more fine-grained discriminative knowledge of the downstream data. ProtoReg substantially improves fine-grained transfer learning performance on multiple benchmarks. Comprehensive experimental results demonstrate that ProtoReg effectively enhances learning fine-grained discriminative information.

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

# A APPENDIX

## A.1 PSEUDO CODE

---

**Algorithm 1:** PyTorch-style Pseudocode for ProtoReg

```
# mb: class-wise memory banks
# cross-entropy: cross-entropy loss function
# aggregation: aggregation loss function
# separation: separation loss function
# lambda_aggr, lambda_sep: coefficients of the aggregation and separation
    loss

# initialize class-discriminative prototypes
if protoreg_self:
   with torch.no_grad():
      model.eval()
      for inputs, labels in loader:
         features, _ = model(inputs, labels)
         mb.enqueue(features, labels)
      model.set_prototypes(mb)
      mb.flush()
elif protoreg_lp:
   model.freeze_backbone()
   for epoch in range(N):
      for inputs, labels in loader:
         _, outputs = model(inputs, labels)
         loss_ce = cross-entropy(outputs, labels)
         loss.backward()
         optimizer.step()
      if no_improvement:
         break
   model.set_prototypes(model.linear_classifier)
   model.unfreeze_backbone()

# fine-tune for M epochs
model.train()
for epoch in range(M):
   for inputs, labels in loader:
      features, outputs = model(inputs, labels)

      # loss calculation
      loss_ce = cross-entropy(outputs, labels)
      loss_aggr = aggregation(features, labels, model.prototypes)
      loss_sep = separation(features, labels, model.prototypes)
      loss = loss_ce + lambda_aggr * loss_aggr + lambda_sep * loss_sep

      # optimization
      loss.backward()
      optimizer.step()

      # enqueue features for prototype refinement
      if protoreg_self:
         mb.enqueue(features.clone().detach(), labels.clone().detach())
   # prototype refinement after each epoch
   if protoreg_self:
      model.set_prototypes(mb)
      mb.flush()
```

---

## A.2 ADDITIONAL DATASET AND IMPLEMENTATION DETAILS

**FGVC Aircraft** includes 100 aircraft model variants with 10,000 images in total. The aircraft model variants are organized hierarchically into 30 manufacturers and 70 families. The dataset is split into three equal parts for training, validation, and testing.

Table 6: The number of samples for the six benchmarks across four different sampling rates. In case the validation set is not available in the benchmark, we randomly selected 20% of the training data to construct the validation data.

| Dataset | Sampling rate | Train | Validation | Test |
|---------|---------------|-------|------------|------|
| FGVC Aircraft | 15%
30%
50%
100% | 500
1,000
1,634
3,334 | 3,333 | 3,333 |
| Stanford Cars | 15%
30%
50%
100% | 961
1,936
3,261
6,509 | 1,635 | 8,041 |
| CUB-200-2011 | 15%
30%
50%
100% | 794
1,400
2,400
4,794 | 1,200 | 5,794 |
| NABirds | 15%
30%
50%
100% | 2,836
5,703
9,563
19,149 | 4,780 | 24,633 |
| Caltech101 | 15%
30%
50%
100% | 650
1,302
2,169
4,338 | 2,169 | 2,170 |
| CIFAR100 | 15%
30%
50%
100% | 6,000
12,000
20,000
40,000 | 10,000 | 10,000 |

**Stanford Cars** is a dataset comprising 196 different car models. The dataset consists of 16,185 images and is split into equal sizes to construct training and test sets. Therefore, each class of training data contained approximately 40 images of cars.

**CUB-200-2011** is a dataset comprising 200 different bird species. It contains a total of 11,788 images: 5,994 images were used for training, while the remaining 5,794 images were used for testing.

**NABirds** encompasses 555 distinct bird species native to North America. It consists of a total of 48,562 images, with 23,929 of these images utilized in the training, and the remaining 24,633 images used for testing.

**Caltech101** is a dataset that contains a diverse collection of images belonging to 101 different categories, including animals, vehicles, household items, and more. It consists of a total of 8,677 images.

**CIFAR100** contains 100 different object classes, each with 600 images, resulting in a total of 60,000 images. The dataset consists of 50,000 training images and 10,000 test images.

Table 6 shows the number of samples we used in four different sampling rate settings.

For data augmentation, we augmented training data with a random resized crop of 224x224 and a random horizontal flip. Validation and test data were first resized to 256x256 and then resized to 224x224 with center cropping. When we initialize prototypes from pre-trained features, we did not use data augmentation to get the representative vector from samples not affected by data augmentation. Each model was optimized using stochastic gradient descent with Nesterov momentum and cosine learning rate scheduling (Loshchilov & Hutter, 2016) over 100 epochs, starting from an initial learning rate of 0.001, except for the linear classifier $\mathcal{G}_\phi$. We used a 10 times larger learning rate for $\mathcal{G}_\phi$, following the settings of prior studies (You et al., 2020; Yosinski et al., 2014). We used a batch size of 64 for all the benchmarks. We used a momentum of 0.9 and weight de-

Table 7: The overall performance of the test accuracy (%) on Caltech101 and CIFAR100. The network is ResNet-50 pre-trained on ImageNet-1K. The baseline is fine-tuning with cross-entropy loss (CE). The compared methods are L2-SP (Xuhong et al., 2018), BSS (Chen et al., 2019), SN (Kou et al., 2020), Co-tuning (You et al., 2020), LP-FT (Kumar et al., 2022), Robust FT (Xiao et al., 2023), and DR-Tune (Zhou et al., 2023).

| Dataset | Method | Sampling rate | | | |
| | | 15% | 30% | 50% | 100% |
|---------|--------|------|------|------|------|
| Caltech101 | CE | 86.54 | 90.55 | 92.26 | 94.29 |
| | L2-SP | 87.28 | 91.06 | 92.67 | 94.24 |
| | BSS | 87.65 | 91.75 | 92.86 | 94.42 |
| | SN | 86.96 | 90.97 | 93.09 | 94.56 |
| | Co-tuning | 87.05 | 90.42 | 92.58 | 94.47 |
| | LP-FT | 86.50 | 91.01 | 93.04 | 94.70 |
| | Robust FT | 86.18 | 90.65 | 93.04 | 93.96 |
| | DR-Tune | 87.60 | 91.29 | 92.90 | 94.75 |
| | ProtoReg (self) | **88.76** | **92.21** | **93.92** | **94.98** |
| | ProtoReg (LP) | 88.57 | 92.17 | **93.92** | 94.89 |
| CIFAR100 | CE | 72.19 | 76.95 | 79.95 | 83.20 |
| | L2-SP | 71.64 | 77.12 | 79.87 | 83.25 |
| | BSS | 71.94 | 77.30 | 80.10 | 83.48 |
| | SN | 71.14 | 76.26 | 79.49 | 82.58 |
| | Co-tuning | 68.26 | 75.25 | 78.20 | 82.44 |
| | LP-FT | 71.25 | 75.99 | 79.45 | 82.64 |
| | Robust FT | 71.92 | 76.12 | 79.54 | 83.09 |
| | DR-Tune | 72.47 | 77.23 | 80.18 | 83.53 |
| | ProtoReg (self) | **73.58** | **78.09** | 80.16 | **83.81** |
| | ProtoReg (LP) | 73.51 | 77.87 | **80.58** | 83.53 |

cay of 0.0001 for the stochastic gradient descent optimizer. Each model was trained using one NVIDIA A40 GPU. We note that our results for the baseline and compared methods are different from the prior study (You et al., 2020) because the number of training samples is different from our implementation. The hyper-parameters were searched for $\lambda_{\text{aggr}} \in \{25, 20, 15, 10, 5, 1, 0\}$ and $\lambda_{\text{sep}} \in \{5, 3, 1, 0.1, 0.01, 0\}$ in ProtoReg (self), and $\lambda_{\text{aggr}} \in \{100, 80, 60, 40, 20\}$ and $\lambda_{\text{sep}} \in \{100, 80, 60, 40, 20\}$ in ProtoReg (LP) using the validation accuracy.

# B    ADDITIONAL EXPERIMENTAL RESULTS

## B.1    TRANSFER LEARNING ON GENERAL IMAGE DATASET

Table 7 presents the transfer learning performance on more general datasets, Caltech101 and CIFAR100. ProtoReg demonstrates superior performance compared to the other methods on both of these datasets.

## B.2    IMPACT OF THE AGGREGATION AND SEPARATION STRENGTHS

In this subsection, we provide the hyper-parameter search results for the three benchmarks across the four sampling rates. Figure 7, 8, and 9 show the test accuracies for FCVC Aircraft, Stanford Cars, and CUB-200-2011 across different sampling rates and hyper-parameter combinations, respectively. The leftmost bar in each subfigure, where $\lambda_{\text{aggr}} = 0$ and $\lambda_{\text{sep}} = 0$, corresponds to the test accuracy obtained when fine-tuning using only cross-entropy loss without employing ProtoReg. Across all three benchmarks, we consistently observed that increasing the $\lambda_{\text{aggr}}$ leads to a continuous improvement in test accuracy. This implies that constraining the pre-trained knowledge that the model utilizes during fine-tuning via $\mathcal{L}_{\text{aggr}}$ plays a significant role in improving the performance of the fine-grained downstream tasks.

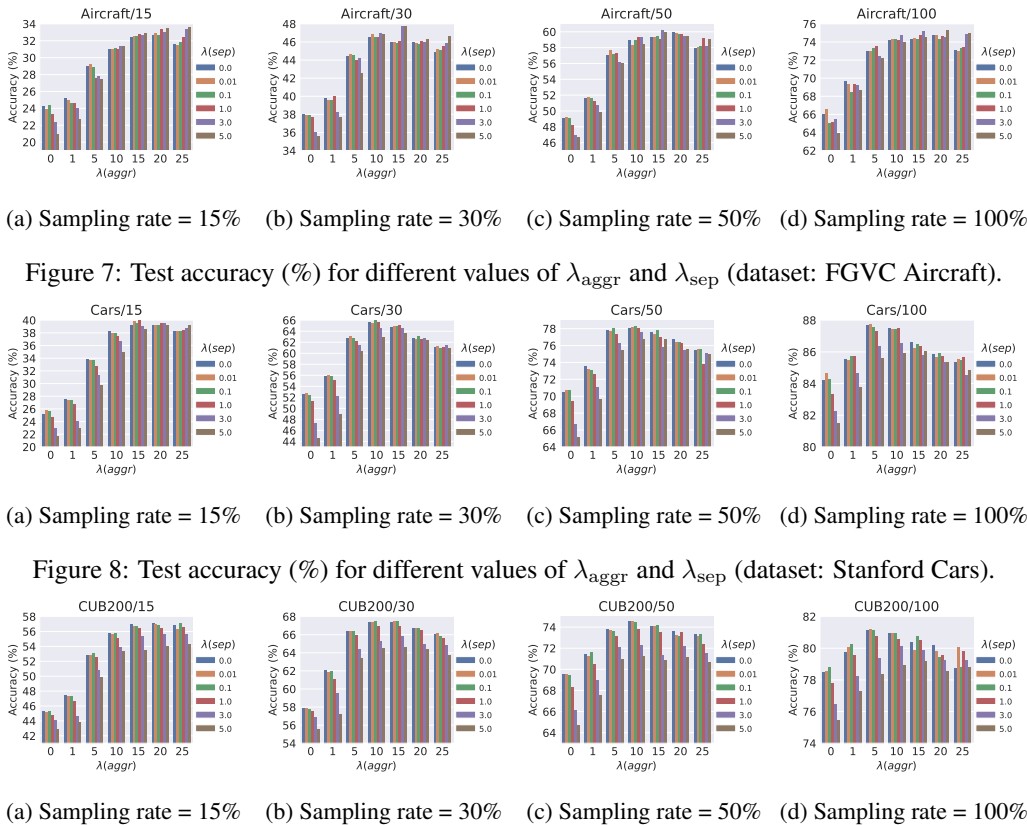

(a) Sampling rate = 15%     (b) Sampling rate = 30%     (c) Sampling rate = 50%     (d) Sampling rate = 100%

Figure 7: Test accuracy (%) for different values of $\lambda_{\text{aggr}}$ and $\lambda_{\text{sep}}$ (dataset: FGVC Aircraft).

(a) Sampling rate = 15%     (b) Sampling rate = 30%     (c) Sampling rate = 50%     (d) Sampling rate = 100%

Figure 8: Test accuracy (%) for different values of $\lambda_{\text{aggr}}$ and $\lambda_{\text{sep}}$ (dataset: Stanford Cars).

(a) Sampling rate = 15%     (b) Sampling rate = 30%     (c) Sampling rate = 50%     (d) Sampling rate = 100%

Figure 9: Test accuracy (%) for different values of $\lambda_{\text{aggr}}$ and $\lambda_{\text{sep}}$ (dataset: CUB-200-2011).

Furthermore, the selection of an appropriate $\lambda_{\text{aggr}}$ is affected by the size of the dataset. In particular, in the sampling rate of 15% (Figure 7a, 8a, and 9a), the test accuracy continues to increase for a large value of $\lambda_{\text{aggr}}$ than that of the higher sampling rate cases. This suggests that in a scenario with limited data, where the model is more susceptible to shortcut learning, it becomes more significant to restrict the impact of pre-trained knowledge that is irrelevant to the specific fine-grained downstream task.

Additionally, we have found that increasing the $\lambda_{\text{sep}}$ can occasionally lead to a substantial decrease in test accuracy. Specifically, when using a large $\lambda_{\text{sep}}$ of 5 in the Stanford Cars and CUB-200-2011 benchmarks, the decline in accuracy was more significant compared to not using $\lambda_{\text{sep}}$ at all. Furthermore, this phenomenon was more noticeable when $\lambda_{\text{aggr}}$ had a smaller value.

## B.3 EFFECTS ON SHORTCUT LEARNING

Figure 10 presents the GradCAM (Selvaraju et al., 2017) visualizations obtained by evaluating the model trained on the CUB-200-2011 using the Waterbirds test data. When fine-tuning was performed using only cross-entropy loss, the model exhibited a tendency to focus on the changed background, rather than the fine-grained discriminative features of the object. Specifically, in layer 4, the model made predictions based on the waves present in the sea. Additionally, layer 2 concentrated on noisy patterns appearing in the surrounding background. In contrast, the model fine-tuned with ProtoReg clearly focused on the bird object. Even in the lower layers, the model exhibited a strong emphasis on the discriminative features of the bird rather than being biased towards the surrounding background.

Table 8 demonstrates that when fine-tuning the model only with cross-entropy loss, shortcut learning occurs from early in the fine-tuning process, hindering the learning of discriminative features required for fine-grained downstream tasks. Specifically, after 5 epochs of training, the training

Table 8: Shortcut learning in the early stage of fine-tuning. The CUB-200-2011 is used as the in-distribution (ID) data, while the out-of-distribution (OOD) data is Waterbirds. The Gen. err. represents the generalization error, indicating the disparity between training and validation accuracy. ↓ denotes the accuracy drop rate on the OOD test data compared to the validation ID accuracy.

| Method | Train (ID) | Validation (ID) | Gen. err. | Test (OOD) | ↓ |
|---|---|---|---|---|---|
| CE (epoch=5) | **71.26** | **68.92** | 2.34 | 51.59 | 25 |
| ProtoReg (epoch=5) | 62.97 | 65.25 | **-2.28** | **52.09** | 20 |
| CE (epoch=10) | **86.47** | **75.67** | 10.80 | 58.99 | 22 |
| ProtoReg (epoch=10) | 78.61 | 74.75 | **3.86** | **62.29** | 17 |
| CE (epoch=100) | **98.56** | 79.25 | 19.31 | 60.2 | 24 |
| ProtoReg (epoch=100) | 97.19 | **81.08** | **16.11** | **70.3** | 13 |

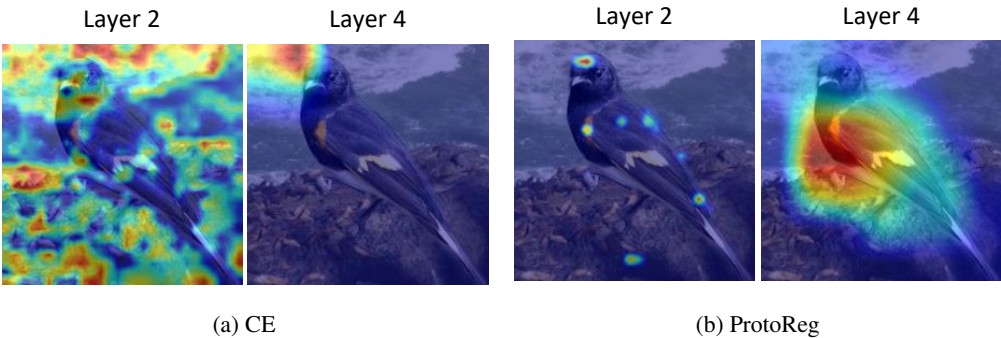

(a) CE  (b) ProtoReg

Figure 10: GradCAM visualizations when evaluating the model trained on the CUB-200-2011 training dataset on a test data sample of Waterbirds.

accuracy is higher in CE, but ProtoReg exhibits lower generalization error and performs better in OOD. This trend persists even after 10 epochs of training, and notably, ProtoReg demonstrates a greater improvement in both generalization error and OOD test accuracy compared to CE.

## B.4 EFFECTS ON FINE-GRAINED CLASSIFICATION

Table 9 compares the classification performance of CE and ProtoReg on more fine-grained classes within the Aircraft benchmark. The four classes presented in Table 9 are aircraft variants manufactured by Boeing, sharing similar characteristics and exhibiting more fine-grained features within the Aircraft dataset. Figure 11 shows the examples of the four fine-grained classes.

When the pre-trained model is fine-tuned with CE, these specific classes demonstrate an average accuracy of 51.07%, which is approximately 15%p lower than the overall class accuracy of 66.01%. In contrast, fine-tuning with ProtoReg yields an average accuracy of 69.14%, showing an increase of 18.07%p compared to CE. Moreover, this accuracy is only around 6%p lower than the overall class accuracy of 75.25%. These results indicate that ProtoReg successfully captures more fine-grained and subtle features, leading to superior classification performance even in more fine-grained classes.

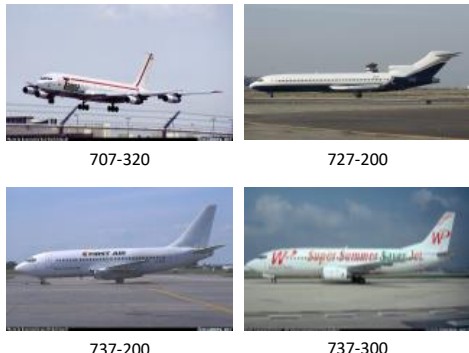

Figure 11: Examples of the four more fine-grained classes within FGVC Aircraft.

Table 9: Validation accuracy (%) on more fine-grained classes within the Aircraft dataset. The four fine-grained classes (707-320, 727-200, 737-200, and 737-300) are aircraft variants from the same manufacturer, i.e., Boeing. @1 and @5 denote the top-1 and top-5 accuracy, respectively.

| Method | Class 707-320 | | 727-200 | | 737-200 | | 737-300 | | Average | |
|---|---|---|---|---|---|---|---|---|---|---|
| | @1 | @5 | @1 | @5 | @1 | @5 | @1 | @5 | @1 | @5 |
| CE | 57.58 | 90.91 | 58.82 | 88.24 | 66.67 | 81.82 | 21.21 | 78.79 | 51.07 | 84.94 |
| ProtoReg | 72.73 | 90.91 | 73.53 | 94.12 | 84.85 | 93.94 | 45.46 | 93.94 | 69.14 | 93.23 |
| $\Delta$ | 15.15 | 0.00 | 14.71 | 5.88 | 18.18 | 12.12 | 24.25 | 15.15 | **18.07** | **8.29** |

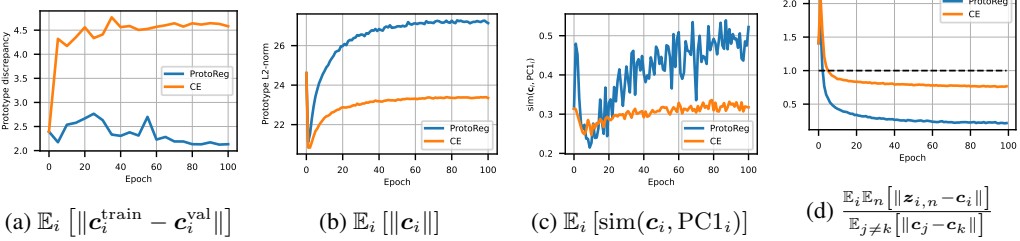

(a) $\mathbb{E}_i\left[\|\boldsymbol{c}_i^{\text{train}} - \boldsymbol{c}_i^{\text{val}}\|\right]$  (b) $\mathbb{E}_i\left[\|\boldsymbol{c}_i\|\right]$  (c) $\mathbb{E}_i\left[\text{sim}(\boldsymbol{c}_i, \text{PC1}_i)\right]$  (d) $\frac{\mathbb{E}_i\mathbb{E}_n\left[\|\boldsymbol{z}_{i,n} - \boldsymbol{c}_i\|\right]}{\mathbb{E}_{j\neq k}\left[\|\boldsymbol{c}_j - \boldsymbol{c}_k\|\right]}$

Figure 12: (a) Expected L2 distance between the training and validation prototype of the same class. (b) Expected L2-norm of the prototype. (c) Expected cosine similarity between prototype and the first principal component of the corresponding class features. (d) Change in the ratio between the distance of a prototype to its corresponding intra-features and the distance between prototypes when the model is fine-tuned using only cross-entropy loss (CE) or jointly with ProtoReg.

### B.5 EFFECTS ON PROTOTYPES

The impact of ProtoReg on the features and prototypes is demonstrated in Figure 12. Figure 12a shows the expected L2 distance between the training and validation prototype of the same class. A high distance indicates a discrepancy between the training and validation prototype, meaning that the prototype is over-fitted to the training data, thereby not appropriately representing the class. When fine-tuning with cross-entropy, the discrepancy increases, especially during the early stage. In contrast, the prototypes regularized with ProtoReg are much less susceptible to discrepancies, and the discrepancy is decreased during the fine-tuning. These findings demonstrate that ProtoReg mitigates the bias towards certain irrelevant features during training, enabling the training prototypes and validation prototypes of each class to better encompass the representative characteristics of the class. This effect is particularly crucial in the early stages of training.

Figure 12b demonstrates that as fine-tuning progresses, ProtoReg exhibits an increasing expected norm of prototypes compared to CE. The $\mathcal{L}_{\text{aggr}}$ of ProtoReg enhances the similarity between features and their corresponding prototypes, causing the angle between the features and prototypes to diminish, thereby constraining the feature representation space. Instead, the increasing norm can be seen as a means for gradually acquiring a larger space in which the features can represent fine-grained information. Figure 12c shows the expected cosine similarity between the prototype and the first principal component of the corresponding class features. The first principal component of class $i$, denoted as $\text{PC1}_i$, is computed through principal component analysis (Jolliffe, 2002) applied to the features of class $i$, i.e., $\{\boldsymbol{z}_{i,n}\}_{n=1}^{N_i^{\text{train}}}$. When using ProtoReg, the prototypes become more aligned with their corresponding principal components, which suggests that the prototypes are oriented toward the direction that contains more information about the class.

Furthermore, Figure 12d illustrates the ratio between $\mathbb{E}_i\mathbb{E}_n\left[\|\boldsymbol{z}_{i,n} - \boldsymbol{c}_i\|\right]$, i.e., the expected Euclidean distance between the features and their corresponding prototype, and $\mathbb{E}_{j\neq k}\left[\|\boldsymbol{c}_j - \boldsymbol{c}_k\|\right]$, i.e., the expected Euclidean distance between different prototypes. Here $\boldsymbol{z}_{i,n}$ denotes the penultimate feature of $\boldsymbol{x}_{i,n}$, i.e., $\boldsymbol{z}_{i,n} = \mathcal{F}_{\boldsymbol{\theta}}(\boldsymbol{x}_{i,n})$. We found that $\mathbb{E}_i\mathbb{E}_n\left[\|\boldsymbol{z}_{i,n} - \boldsymbol{c}_i\|\right] > \mathbb{E}_{j\neq k}\left[\|\boldsymbol{c}_j - \boldsymbol{c}_k\|\right]$ holds in the initial state, indicating a high intra-class variance compared to the distance between prototypes

in the pre-trained feature space for the fine-grained downstream task. As fine-tuning progresses, $\mathcal{L}_{\text{aggr}}$ decreases the intra-class variance, while $\mathcal{L}_{\text{sep}}$ increases the distance between prototypes. Consequently, this leads to a decreased ratio between $\mathbb{E}_i \mathbb{E}_n \left[\|z_{i,n} - c_i\|\right]$ and $\mathbb{E}_{j \neq k} \left[\|c_j - c_k\|\right]$ compared to when using only CE.

## B.6 Learning Curves

Figure 13 represents the learning curves for training and validation accuracy on the FGVC Aircraft dataset. The pre-trained model is fine-tuned for 400 epochs, with ProtoReg employing values of $\lambda_{\text{aggr}}$ and $\lambda_{\text{sep}}$ set to 40 and 5, respectively. Fine-tuning with only the CE results in a rapid convergence of the training accuracy, typically within approximately 50 epochs, and a plateau in validation accuracy. In contrast, adopting the ProtoReg regularizer leads to a slower convergence of the training accuracy, yet the validation accuracy continues to improve gradually. This observation suggests that the regularized features adapt at a slower pace, allowing them to acquire

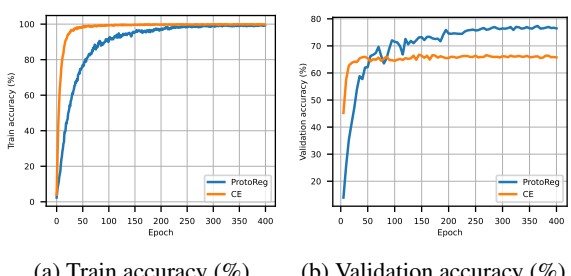

(a) Train accuracy (%)    (b) Validation accuracy (%)

Figure 13: Learning curves of the training and validation accuracy.

more generalizable characteristics over time.

## B.7 Ablation Study on Prototype Self-refinement

Table 10 presents the results of an ablation study on the prototype self-refinement approach. ProtoReg utilizes class-wise memory banks, where the computed features from each mini-batch are enqueued into their corresponding memory banks. At the end of each epoch, these features are utilized to update the prototypes. As alternatives, we compared the per-step method, where the prototypes are updated after each iteration step by recomputing the entire feature set, and the per-epoch method, where the prototypes are updated after each epoch by

Table 10: Test accuracy (%) for different prototype self-refinement methods.

| Refinement | Accuracy (%) | Relative Time |
|---|---|---|
| Per-epoch | 73.99 | 1.2 |
| Per-step | 74.26 | 22.2 |
| Ours | 75.25 | 1 |

recomputing the entire feature set. Our approach demonstrated higher accuracy compared to the comparative methods, with improvements of 1.26%p and 0.99%p respectively. Additionally, our method had the advantage of faster training speed since there was no need to recompute the features at each update.

## B.8 Effects on Model Calibration

To evaluate the calibration of the model, we computed the expected calibration error (ECE) (Naeini et al., 2015) on the validation dataset. ECE quantifies the discrepancy between the predicted confidence of a model and its true accuracy. The results in Table 11 demonstrate that the model trained with ProtoReg achieved a significantly lower ECE of 0.038, compared to the ECE of 0.115 obtained by fine-tuning with cross-entropy loss. This suggests that ProtoReg is effective in improving the calibration of the model.

Table 11: Expected Calibration Error (ECE) on the validation data.

| Method | ECE $\downarrow$ |
|---|---|
| CE | 0.115 |
| ProtoReg | 0.038 |

## B.9 Additional Results on Image Retrieval

Figures 14 and 15 present additional image retrieval results on CUB-200-2011 validation samples. In the left results of Figure 14, the features obtained from the pre-trained model exhibit a bias towards the background (in this case, the ground). With CE, there is

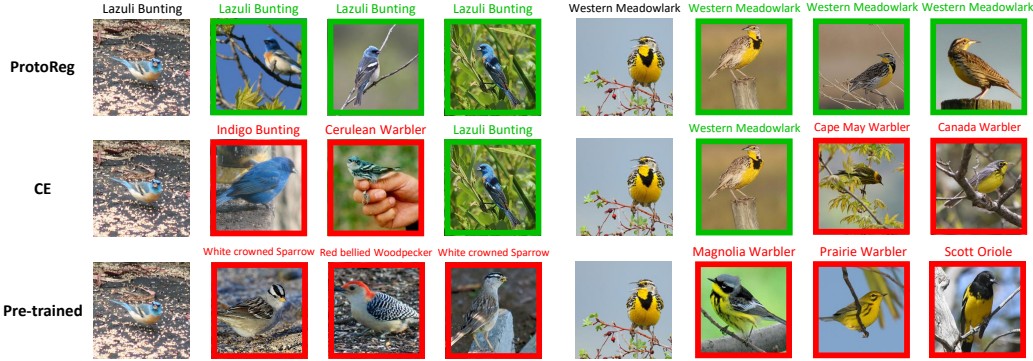

Figure 14: Qualitative evaluation of image retrieval using the validation samples from the CUB-200-2011 dataset. Ground truth: Lazuli Bunting (left), Western Meadowlark (right).

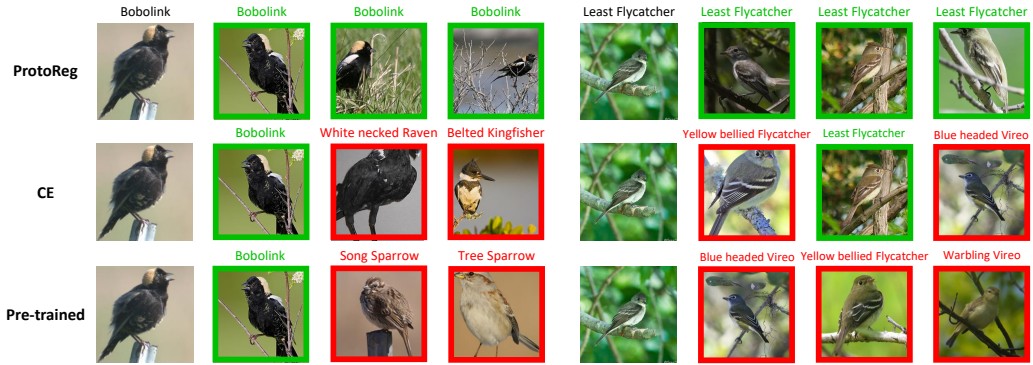

Figure 15: Qualitative evaluation of image retrieval using the validation samples from the CUB-200-2011 dataset. Ground truth: Bobolink (left), Least Flycatcher (right).

a bias towards the color of the birds, while ProtoReg successfully retrieves birds of the same class. In the right results of Figure 14, the pre-trained model shows a bias towards the yellow color. CE still exhibits a bias towards color, resulting in the retrieval of yellow birds without the distinctive characteristics of the ground truth.

In the left results of Figure 15, when retrieved from the pre-trained model, samples from different classes but with similar backgrounds are retrieved. With CE, there is a retrieval of samples with similar colors. On the other hand, ProtoReg successfully retrieves samples from the same class regardless of the background. In the right results of Figure 15, a similar pattern can be observed. The pre-trained model and CE retrieve birds in similar poses perched on branches, while ProtoReg captures the distinctive features of the corresponding class more effectively.

## B.10 QUANTITATIVE RESULTS ON IMAGE RETRIEVAL

In addition to the qualitative visual samples, we present quantitative metrics related to image retrieval: mAP @K and recall @K. mAP calculates the average precision of search results, with mAP @K focusing on the average precision within the top K results. A higher mAP @K indicates increased accuracy and consistency in retrieval. Conversely, recall @K measures the ability to retrieve actual positive instances within the top K results, with higher values indicating better capability in finding relevant items. Table 12 demonstrates that using ProtoReg leads to a significant improvement in both metrics compared to CE.

Table 12: mAP @K and recall @K on image retrieval.

| Method | K = 1 | | K = 3 | | K = 5 | |
|---|---|---|---|---|---|---|
| | mAP | Recall | mAP | Recall | mAP | Recall |
| CE | 0.640 | 0.640 | 0.701 | 0.548 | 0.690 | 0.461 |
| ProtoReg (self) | 0.699 | 0.699 | 0.754 | 0.631 | 0.742 | 0.553 |
| ProtoReg (LP) | **0.738** | **0.738** | **0.787** | **0.689** | **0.781** | **0.612** |

Table 13: Winning ratio in classifying OOD test data.

(a) ProtoReg (self) vs CE

| K | ProtoReg (self) win | CE win | Both wrong | Both correct |
|---|---|---|---|---|
| 1 | **11.65** | 2.71 | 28.41 | 57.23 |
| 3 | **9.04** | 1.42 | 14.34 | 75.20 |
| 5 | **7.77** | 0.86 | 10.32 | 81.05 |

(b) ProtoReg (LP) vs CE

| K | ProtoReg (LP) win | CE win | Both wrong | Both correct |
|---|---|---|---|---|
| 1 | **16.88** | 2.76 | 23.18 | 57.18 |
| 3 | **11.75** | 1.40 | 11.63 | 75.22 |
| 5 | **10.22** | 1.16 | 7.87 | 80.76 |

## B.11 QUANTITATIVE RESULTS ON LEARNING CLASS-DISCRIMINATIVE INFORMATION

In Table 2 and Figure 3, we demonstrated that ProtoReg robustly classifies OOD test data where the non-discriminative and coarse information is disrupted, indicating that the regularized model effectively focuses on the fine-grained discriminative information. To further verify the effective learning of discriminative information in ProtoReg, we further provide a quantitative analysis of whether CE tends to misclassify the misclassified samples on ProtoReg. For Waterbirds test data, we compared methods A and B by calculating ratios for four cases: 1) A correct and B incorrect, 2) A incorrect and B correct, 3) both incorrect, and 4) both correct. Table 13 presents the results for top-K predictions based on the values of K. ProtoReg (self) and ProtoReg (LP) significantly outperform CE, winning by 11.65% and 16.88%, respectively. A noteworthy observation is that when ProtoReg makes a mistake, CE tends to make the same mistake in almost all cases. Out of the 31.12% errors made by ProtoReg (self), only 2.71% are correctly predicted by CE, while the remaining 28.41% are also misclassified for CE.

## B.12 COMPUTAIONAL OVERHEAD

As the similarity computation can be implemented with a single matrix multiplication, ProtoReg results in minimal computational overhead. Specifically, using l2-normalized feature matrix $F \in \mathbb{R}^{B \times d}$ and l2-normalized prototypes matrix $C \in \mathbb{R}^{K \times d}$ (where B is batch size, d is feature dimension, and K is the number of classes), the cosine similarity between each sample and prototype is computed in a single matrix multiplication via $FC^T \in \mathbb{R}^{B \times K}$. The element (i, j) indicates the cosine similarity between the i-th sample and the j-th class pro-

Table 14: Computation overhead compared to CE.

| Method | Relative Time |
|---|---|
| CE | 1 |
| ProtoReg (self) | 1.03 |
| ProtoReg (LP) | 1.12 |

totype. Table 14 demonstrates the relative time comparison against CE when experiments are conducted on the same device. ProtoReg (self) has a negligible computational overhead of only 3%, while ProtoReg (LP) has a 12% overhead due to the linear probing process for prototype initialization. In both cases, the overhead is sufficiently low, considering the significant performance improvement they offer.

