# OpenReview forum: "ProtoReg: Prioritizing Discriminative Information for Fine-grained Transfer Learning"
_ICLR.cc/2024/Conference — Submitted to ICLR 2024_

### Official Review · Reviewer_bkLU · 2023-10-24

**Soundness:** 3 good
**Presentation:** 2 fair
**Contribution:** 3 good
**Rating:** 6
**Confidence:** 5

**Summary:**

This paper proposes a prioritizing discriminative information method to handle the granularity gap challenge in fine-grained transfer learning. It first computes class-discriminative prototypes and then refines the prototypes to gradually capture more fine-grained details as well as aggregate features with corresponding prototypes.

**Strengths:**

1.	This paper points out a key challenge in fine-grained transfer learning, the granularity gap challenge, and proposes a novel ProtoReg approach to address this.
2.	Experimental results demonstrate the effectiveness of the proposed ProtoReg method on both inter-dataset and intra-dataset transfer learning.
3.	Delicate visualization results are provided as clear illustrations of the motivation and effectiveness of the proposed method.
4.	A valid appendix is provided to improve the completeness of this article.

**Weaknesses:**

1.	More details of ablation studies should be provided to further clarify the effectiveness of the proposed method. For example, the experiments with configurations “L_ce + L_aggr + L_step”, “L_ce + L_step + Refine” should be conducted to demonstrate the effectiveness of L_step when applied under different conditions.
2.	Some details of the proposed method should be stated more clearly. For example, in Equ (2), how \phi is defined and how the “argmax” value is computed with respect to the classification accuracy should be explicitly declared.

**Questions:**

1.	Can ProtoReg outperform SOTA methods on other downstream tasks of transfer learning, such as semantic segmentation?

---

> ### Author Response · Authors · 2023-11-17
> **Response to Reviewer bkLU**
>
> > Q1: More details of ablation studies should be provided to further clarify the effectiveness of the proposed method. For example, the experiments with configurations “L_ce + L_aggr + L_step”, “L_ce + L_step + Refine” should be conducted to demonstrate the effectiveness of L_step when applied under different conditions.
>
> A1: Thank you for your valuable feedback. We conducted additional ablation studies based on your comments and incorporated the corresponding results into Table 4, highlighted in red in the revised manuscript.
>
> |$\mathcal{L}_{ce}$|$\mathcal{L}_{aggr}$|$\mathcal{L}_{sep}$|Refine|Accuracy (%)|
> | --- | --- | --- | --- | --- |
> |O|O|O||71.23|
> |O||O|O|63.79|
>
>
> ---
>
>
> > Q2: Some details of the proposed method should be stated more clearly. For example, in Equ (2), how \phi is defined and how the “argmax” value is computed with respect to the classification accuracy should be explicitly declared.
>
> A2: Thank you for your comments. We have revised the manuscript (section 4.2) to clarify the methodology based on your comments, and the modified contents are indicated in red. In Equation (2), $\phi$ is randomly initialized. It should be noted that when initializing prototypes in Equation (2), it is independent of $\phi$ since only the pre-trained feature extractor $\mathcal{F}_{\theta}$ is used.
> Equation (3) with "argmax" is employed when initializing prototypes using linear classifier weights. Examining Equation (3), "argmax" corresponds to $\phi$ that maximizes validation accuracy on downstream data. Specifically, we train a linear classifier for downstream training data with a frozen feature extractor, measuring validation accuracy every certain epoch (in our case, every 5 epochs for computational efficiency). As validation accuracy typically decreases after increasing (suggesting the start of overfitting on training data), we apply early stopping when validation accuracy starts to decrease and use the corresponding $\phi$ as the argmax. This approach ensures prototype initialization that incorporates discriminative information of the pre-trained feature extractor without overfitting to downstream training data.
>
>
> ---
>
>
> > Q3. Can ProtoReg outperform SOTA methods on other downstream tasks of transfer learning, such as semantic segmentation?
>
>
> A3: In our research, we consider a classification task as the downstream task for transfer learning. Given that the semantic segmentation task involves multiple classes within a single image sample, it is not suitable for our setting. Instead, ProtoReg demonstrates notable performance improvement not limited to fine-grained classification but also on more general datasets like Caltech101 and CIFAR100 (see Table 7 in the Appendix).

---

### Official Review · Reviewer_BWuH · 2023-10-31

**Soundness:** 2 fair
**Presentation:** 3 good
**Contribution:** 1 poor
**Rating:** 5
**Confidence:** 4

**Summary:**

The authors proposed prototype-based regularization losses to improve transfer learning for fine-grained classification. To be specific, adaptively evolving prototypes for each class are first introduced. The aggregation loss, which aggregates features with their corresponding prototypes, and the separation loss, which enforces features away from prototypes of different classes, are jointly optimized with the cross-entropy loss for fine-tuning. Comprehensive experiments are conducted to validate the efficacy of the proposed method.

**Strengths:**

1. The paper is well-written, making it easy to follow.
2. Sufficient experimental studies are provided.

**Weaknesses:**

1. The idea of prototype-based loss is not novel for dealing with the limited data scenario. Such an idea could be tracked back to the early few-shot research [1]. The proposed aggregation loss and the separation loss are same as the Eq. (2) in [1].
2. The challenge of fine-grained classification I think lies in its limited data for each class, tens of samples for each class in most benchmark datasets (Table 6 A.2). For CIFAR100, hundreds of samples for each class, the proposed method achieves insignificant perform gains compared to CE (Table 7). Therefore, I think the few-shot works are related and should be included in the literature review.
3. Some experimental results seem unfair. For the results in Figure 1 and Figure 3, the case where ProtoReg succeeds while others fails is picked up. A question arises here: whether the samples that ProtoReg mis-classify are mis-classified by CE? If so, the extra mis-classification by CE would indicate its deficiency. Otherwise, ProtoReg and CE possibly have different shortcuts. The results presented should be more statistical.

Minor:
1. The pseudo-code did not include the refinement for the prototypes based on linear classifier weights.

References:
[1] Snell, J., Swersky, K., & Zemel, R. (2017). Prototypical networks for few-shot learning. Advances in neural information processing systems, 30.

**Questions:**

See the above weakness.

---

> ### Author Response · Authors · 2023-11-17
> **Response to Reviewer BWuH (1/3)**
>
> > Q1: The idea of prototype-based loss is not novel for dealing with the limited data scenario. Such an idea could be tracked back to the early few-shot research [1]. The proposed aggregation loss and the separation loss are same as the Eq. (2) in [1].
>
> A1: Our methodology introduces novelty in the following aspects (This content is identical to the response to the Reviewer Ka1A's question 1. Additionally, in the later sections, we summarized the distinctions from [1] and the resulting effects.):
>
> * **Novel prototype initialization method**: Previous approaches in using prototypes typically adopted class-wise sample mean initialization. We propose initializing prototypes with weights learned from a linear classifier, addressing the lack of discriminative information in fine-grained data. Our approach demonstrates notable performance improvement.
> * **Notable performance enhancement in transfer learning**: Prior works leveraging prototypes have primarily focused on few-shot learning [1, 2] and domain adaptation [3, 4], leaving the effectiveness in transfer learning underexplored. We identify the motivational observation of the granularity gap in fine-grained transfer learning and highlight that prototype-based regularization can be a highly effective solution.
> * **Refinement strategy**: Prior works either did not refine prototypes or required recomputation of the entire feature set, leading to computational inefficiency. We propose a computationally efficient approach that adaptively evolves prototypes to include discriminative information learned during fine-tuning.
> * **Exploration of separated loss components**: Previous works often formulate prototype-based methods using log-likelihood [1, 2] or distance minimization in a metric space [5]. We meticulously explore the disentanglement and the importance of assigning different strengths to each component for the first time (which are discussed in section 5.4.2 and Appendix B.2). The table below further illustrates that our approach, which assigns different strengths to $\lambda_{aggr}$ and $\lambda_{sep}$, significantly enhances performance compared to cases where the both strengths are equal to $\alpha$.
>
> (a) FGVC Aircraft
>
> |$\alpha$| 15%|	30%|	50%|	100%|
> |---|---|---|---|---|
> |1.0|	24.57|	39.96|	51.25|	69.31|
> |5.0|	27.39|	42.60|	56.02|	72.22|
> |10.0|	30.69|	45.37|	56.77|	73.03|
> |15.0|	31.62|	45.64|	57.82|	74.14|
> |20.0|	31.92|	45.49|	56.74|	73.00|
> |25.0|	31.53|	45.31|	56.71|	71.89|
> |Ours|	**33.66**|	**46.83**|	**59.95**|	**75.25**|
>
> (b) Stanford Cars
>
> |$\alpha$| 15%|	30%|	50%|	100%|
> |---|---|---|---|---|
> |1.0|	26.68|	55.01|	72.64|	85.70|
> |5.0|	29.74|	60.44|	75.41|	85.57|
> |10.0|	32.96|	60.35|	74.10|	85.60|
> |15.0|	33.16|	59.57|	74.23|	85.55|
> |20.0|	32.67|	59.00|	73.46|	84.83|
> |25.0|	30.95|	56.25|	70.95|	84.42|
> |Ours|	**39.95**|	**65.91**|	**78.36**|	**87.74**|
>
> (c) CUB-200-2011
>
> |$\alpha$| 15%|	30%|	50%|	100%|
> |---|---|---|---|---|
> |1.0|	46.60|	61.00|	70.47|	79.57|
> |5.0|	49.78|	63.36|	70.95|	78.34|
> |10.0|	50.54|	61.37|	68.30|	77.13|
> |15.0|	49.26|	59.11|	66.67|	76.04|
> |20.0|	47.14|	58.10|	65.93|	75.77|
> |25.0|	46.41|	56.85|	64.67|	74.80|
> |Ours|	**57.09**|	**67.48**|	**74.53**|	**81.22**|
>
> In direct comparison with [1], the differences are evident.
> * [1] lacks prototype refinement and relies on episodic training for few-shot learning, which doesn't align with the transfer learning setting where the entire dataset of all classes is typically utilized at once. Prototype refinement enables adaptation to the changing feature space during fine-tuning and allows for the adaptive evolution of prototypes.
> * In prototype initialization, while [1] uses sample mean, we introduce a novel approach using linear classifier weights to maximize the inclusion of discriminative information. This novel approach allows for the initialization of prototypes that effectively incorporate discriminative information even in fine-grained scenarios.
> * While [1] employs maximum log-likelihood optimization, we demonstrate significant performance improvement by disentangling through aggregation and separation, assigning different strengths to each term. The above experimental results demonstrate that significant performance improvement is achievable by simply separating aggregation and separation and assigning different weights to each.
>
>
> References
> [1] Snell et al., "Prototypical networks for few-shot learning.", NeurIPS 2017
> [2] Hou et al., "A closer look at prototype classifier for few-shot image classification.", NeurIPS 2022
> [3] Pan et al., "Transferrable prototypical networks for unsupervised domain adaptation.", CVPR 2019
> [4] Tanwisuth et al., "A prototype-oriented framework for unsupervised domain adaptation.", NeurIPS 2021
> [5] Yang et al., "Robust classification with convolutional prototype learning.", CVPR 2018

---

> ### Author Response · Authors · 2023-11-17
> **Response to Reviewer BWuH (2/3)**
>
> > Q2: The challenge of fine-grained classification I think lies in its limited data for each class, tens of samples for each class in most benchmark datasets (Table 6 A.2). For CIFAR100, hundreds of samples for each class, the proposed method achieves insignificant perform gains compared to CE (Table 7). Therefore, I think the few-shot works are related and should be included in the literature review.
>
> A2: Thank you for your valuable feedback. However, the challenge in fine-grained classification we addresses goes beyond simply having limited data for each class. To demonstrate this, we equalized the number of samples per class on the FGVC Aircraft and CIFAR100 benchmark datasets (both datasets have a total of 100 classes). Specifically, to align with the fewer data in the FGVC Aircraft dataset (averaging 33 samples per class), we adjusted the sampling rate for CIFAR100 to include an average of 33 samples per class as well.
>
>
> | Methods         | Aircraft | CIFAR100 |
> | --------------- | -------- | -------- |
> | CE              | 66.01    | 66.87    |
> | ProtoReg (self) | 75.25    | 68.12    |
> | ProtoReg (LP)   | 77.89    | 68.39    |
>
> The experiment results reveal that, even in a setting where the number of classes and samples per class are equal, fine-tuning with cross-entropy faces more difficulty and is sub-optimal in the fine-grained dataset. While there is a notable performance improvement by ProtoReg in CIFAR100 as well, the impact is more pronounced in the fine-grained dataset.
>
> Furthermore, few-shot learning works typically experiment with highly limited data scenarios, such as 5-way 1-shot or 5-way 5-shot. In contrast, our transfer learning scenario involves using a much larger number of classes and samples at once, like 100-way 33-shot (Aircraft) or 555-way 35-shot (NABirds). Therefore, we believe comparisons with few-shot learning works are not fair. Even in settings with a low sampling rate, such as 15%, the number of classes is still considerably larger compared to few-shot settings, ranging from 100 to 555.
>
> > Q3. Some experimental results seem unfair. For the results in Figure 1 and Figure 3, the case where ProtoReg succeeds while others fails is picked up. A question arises here: whether the samples that ProtoReg mis-classify are mis-classified by CE? If so, the extra mis-classification by CE would indicate its deficiency. Otherwise, ProtoReg and CE possibly have different shortcuts. The results presented should be more statistical.
>
> A3: In response to your comment, we have conducted additional experiments (related to Figure 1 and Figure 3) that provide more statistical analysis. We added the results in the revised manuscript, highlighted in red (Appendix B.10 and B.11)
>
> [statistical analysis for Figure 1]
> In Figure 1, we included visual results to highlight our motivation. Below, we present quantitative metrics related to image retrieval: mAP @K and Recall @K. mAP calculates the average precision of search results, with mAP @K focusing on the average precision within the top K results. A higher mAP @K indicates increased accuracy and consistency in retrieval. Conversely, recall @K measures the ability to retrieve actual positive instances within the top K results, with higher values indicating better capability in finding relevant items. Experimental results demonstrate that using ProtoReg leads to a significant improvement in both metrics compared to CE.
>
> - mAP @K
>
> |K|CE|ProtoReg (self)|ProtoReg (LP)|
> | --- | --- | --- | --- |
> |1|0.640|0.699|**0.738**|
> |3|0.701|0.754|**0.787**|
> |5|0.690|0.742|**0.781**|
>
>
> - Recall @K
>
> |K|CE|ProtoReg (self)|ProtoReg (LP)
> |---|---|---|---|
> |1|0.640|0.699|**0.738**|
> |3|0.548|0.631|**0.689**|
> |5|0.461|0.553|**0.612**|
>
>
> [statistical analysis for Figure 3]
> For OOD test data (Waterbirds), we compared methodologies A and B by calculating ratios for four cases: 1) A correct and B incorrect, 2) A incorrect and B correct, 3) both incorrect, and 4) both correct. The results for top-K predictions are presented based on the values of K. ProtoReg (self) and ProtoReg (LP) significantly outperform CE, winning by 11.65% and 16.88%, respectively. A noteworthy observation is that when ProtoReg makes a mistake, CE tends to make the same mistake in almost all cases. Out of the 31.12% errors made by ProtoReg (self), only 2.71% are correctly predicted by CE, while the remaining 28.41% are also misclassified for CE. We believe this percentage is sufficiently low to demonstrate CE's deficiency.
>
> - ProtoReg (self) vs CE
>
> | K   | ProtoReg (self) win |CE win| Both wrong | Both correct|
> | --- | -------- | --- | --- | --- |
> | 1   |**11.65**|2.71|28.41|57.23|
> | 3   |**9.04**|1.42|14.34|75.20|
> | 5   |**7.77**|0.86|10.32|81.05|
>
>
> - ProtoReg (LP) vs CE
>
> |K|ProtoReg (LP) win|CE win|Both wrong|Both correct|
> |---|---|---|---|---|
> |1|**16.88**|2.76|23.18|57.18|
> |3|**11.75**|1.40|11.63|75.22|
> |5|**10.22**|1.16|7.87|80.76|

---

> ### Author Response · Authors · 2023-11-17
> **Response to Reviewer BWuH (3/3)**
>
> > Q4. The pseudo-code did not include the refinement for the prototypes based on linear classifier weights.
>
> A4: When refining prototypes initialized based on linear classifier weights, the prototypes are made learnable and updated through backpropagation. Refinement takes place seamlessly after loss.backward() in the optimizer.step(), without the need for additional pseudo-code.

---

> > ### Comment · Reviewer_BWuH · 2023-11-22
> > **I appreciate the authors for their efforts. However, I am not fully convinced by their responses.**
> >
> > 1. The connection between prototypes and weights of a linear classifier has already been explored in [1, 2]. So, the authors highlighted “novel prototype initialization method” and “refinement strategy” are marginal contributions.
> >
> > Also, I do not agree the claim “Prior works leveraging prototypes have primarily focused on few-shot learning [1, 2] and domain adaptation [3, 4], leaving the effectiveness in transfer learning under-explored”. Basically, domain adaptation is a subcategory of transfer learning, and one branch of few-shot learning adopts transfer learning-based techniques [2].
> >
> > 2. I think the presented results, demonstrating the proposed method's notable performance improvement on CIFAR-100 with less data compared to the original CIFAR-100, support my argument. The amount of data indeed introduces a distinct aspect.
> >
> > Yes, few-shot learning works typically involves working with extremely limited data and are supposed to work better in less challenging scenarios with relatively limited data.
> > Therefore, I suggest including these works in the literature review, especially given the fact that the proposed methods are based on prototypes.
> >
> > Overall, I think the work makes marginal contributions, and thus I keep my score.
> >
> >
> > References
> >
> > [1] Snell, J., Swersky, K., & Zemel, R. (2017). Prototypical networks for few-shot learning. Advances in neural information processing systems, 30.
> >
> > [2] Medina, C., Devos, A., & Grossglauser, M. (2020). Self-supervised prototypical transfer learning for few-shot classification. arXiv preprint arXiv:2006.11325.

---

### Official Review · Reviewer_Ka1A · 2023-11-01

**Soundness:** 3 good
**Presentation:** 3 good
**Contribution:** 2 fair
**Rating:** 5
**Confidence:** 3

**Summary:**

This paper focuses on the problem of fine-tuning a pre-trained model for a fine-grained downstream task. The authors point out that there is a granularity gap between the information learned by the pre-trained model and the fined-grained downstream task, and then propose a fine-tuning approach by utilizing prototypes to prioritize the discriminative information. The experimental results show that the method is able to effectively improve the performance of the fine-grained classification tasks.

**Strengths:**

1. The topic this paper pays attention to is important. As fine-tuning a pre-trained model for a particular task has been a standard learning paradigm in computer vision, it is valuable to study how to deal with the case where the downstream task is fine-grained, which has not been studied deeply.
2. The motivation of this paper is clear. I agree with the authors' viewpoint that there is a granularity gap between the level of information learned by the pre-trained model and the semantic details required for a fine-grained downstream task, which is the main problem to solve in fine-trained transfer learning.
3. The proposed method based on prototype aggregation is straightforward and not difficult to realize.
4. The English writing of this paper is generally good.

**Weaknesses:**

1. The novelty of this paper is not so significant. The idea of exploiting prototypes to represent and adjust the feature distribution according to label information from samples has been studied in a lot of works. Although the idea of utilizing such mechanism to fine-tune a pre-trained model for downstream fine-grained tasks is new, I do not recognize significant difference between the key idea of this work and the previous ones. It does has some novelty, but I am not sure whether it is sufficient to meet the criterion of ICLR.
2. Although the motivation of solving the problem of granularity gap is interesting, I do not quite understand how the proposed mechanism with prototypes helps to eliminate such "gap" under the perspective of transfer learning. In more detail, it seems that this method can also be used to train a randomly initialized model for the fine-grained task, because I do not see a mechanism to "preserve" useful information from the original pre-trained model but only recognize the mechanism to make change to the model.
3. A minor problem: the last sentence of the comments of Figure 1 is confusing. Maybe the word "suffer from" is not correctly used.

**Questions:**

1. Considering the 2nd drawback listed above, please explain whether there is a mechanism to discover and  preserve useful information from the pre-trained model.
2. It is a common scene for real data that each class may have multiple prototypes. How do the authors deal with such a case?
3. Please explain why two different prototype updating strategies are used according to their initialization methods? What is the motivation or consideration of their designs, respectively?

**Details Of Ethics Concerns:**

I have no ethics concerns about this paper.

---

> ### Author Response · Authors · 2023-11-17
> **Response to Reviewer Ka1A (1/3)**
>
> > **Q1: The novelty of this paper is not so significant. The idea of exploiting prototypes to represent and adjust the feature distribution according to label information from samples has been studied in a lot of works. Although the idea of utilizing such mechanism to fine-tune a pre-trained model for downstream fine-grained tasks is new, I do not recognize significant difference between the key idea of this work and the previous ones.**
>
> A1: Our methodology introduces novelty in the following aspects:
>
> * **Novel prototype initialization method**: Previous approaches in using prototypes typically adopted class-wise sample mean initialization. We propose initializing prototypes with weights learned from a linear classifier, addressing the lack of discriminative information in fine-grained data. Our approach demonstrates notable performance improvement.
> * **Notable performance enhancement in transfer learning**: Prior works leveraging prototypes have primarily focused on few-shot learning [1, 2] and domain adaptation [3, 4], leaving the effectiveness in transfer learning underexplored. We identify the motivational observation of the granularity gap in fine-grained transfer learning and highlight that prototype-based regularization can be a highly effective solution.
> * **Refinement strategy**: Prior works either did not refine prototypes or required recomputation of the entire feature set, leading to computational inefficiency. We propose a computationally efficient approach that adaptively evolves prototypes to include discriminative information learned during fine-tuning.
> * **Exploration of separated loss components**: Previous works often formulate prototype-based methods using log-likelihood [1, 2] or distance minimization in a metric space [5]. We meticulously explore the disentanglement and the importance of assigning different strengths to each component for the first time (which are discussed in section 5.4.2 and Appendix B.2). The table below further illustrates that our approach, which assigns different strengths to $\lambda_{aggr}$ and $\lambda_{sep}$, significantly enhances performance compared to cases where the both strengths are equal to $\alpha$.
>
> (a) FGVC Aircraft
>
> |$\alpha$| 15%|	30%|	50%|	100%|
> |---|---|---|---|---|
> |1.0|	24.57|	39.96|	51.25|	69.31|
> |5.0|	27.39|	42.60|	56.02|	72.22|
> |10.0|	30.69|	45.37|	56.77|	73.03|
> |15.0|	31.62|	45.64|	57.82|	74.14|
> |20.0|	31.92|	45.49|	56.74|	73.00|
> |25.0|	31.53|	45.31|	56.71|	71.89|
> |Ours|	**33.66**|	**46.83**|	**59.95**|	**75.25**|
>
> (b) Stanford Cars
>
> |$\alpha$| 15%|	30%|	50%|	100%|
> |---|---|---|---|---|
> |1.0|	26.68|	55.01|	72.64|	85.70|
> |5.0|	29.74|	60.44|	75.41|	85.57|
> |10.0|	32.96|	60.35|	74.10|	85.60|
> |15.0|	33.16|	59.57|	74.23|	85.55|
> |20.0|	32.67|	59.00|	73.46|	84.83|
> |25.0|	30.95|	56.25|	70.95|	84.42|
> |Ours|	**39.95**|	**65.91**|	**78.36**|	**87.74**|
>
> (c) CUB-200-2011
>
> |$\alpha$| 15%|	30%|	50%|	100%|
> |---|---|---|---|---|
> |1.0|	46.60|	61.00|	70.47|	79.57|
> |5.0|	49.78|	63.36|	70.95|	78.34|
> |10.0|	50.54|	61.37|	68.30|	77.13|
> |15.0|	49.26|	59.11|	66.67|	76.04|
> |20.0|	47.14|	58.10|	65.93|	75.77|
> |25.0|	46.41|	56.85|	64.67|	74.80|
> |Ours|	**57.09**|	**67.48**|	**74.53**|	**81.22**|
>
>
> References
> [1] Snell et al., "Prototypical networks for few-shot learning.", NeurIPS 2017
> [2] Hou et al., "A closer look at prototype classifier for few-shot image classification.", NeurIPS 2022
> [3] Pan et al., "Transferrable prototypical networks for unsupervised domain adaptation.", CVPR 2019
> [4] Tanwisuth et al., "A prototype-oriented framework for unsupervised domain adaptation.", NeurIPS 2021
> [5] Yang et al., "Robust classification with convolutional prototype learning.", CVPR 2018

---

> > ### Comment · Reviewer_Ka1A · 2023-11-19
> >
> > I have read the responses from the authors and thanks to the authors for their efforts. However, I am sorry that these explanations cannot persuade me to change my rating about this paper.
> >
> > On one side, the novelty of this paper is a little weak compared with the criterion of ICLR in my opinion. I do not understand why the authors assert that prior works either did not refine prototypes in their response. Indeed, most methods based on prototype learning have to adjust the prototypes for learning better feature representation of each class. On the other side, since this method does not intend to preserve information from the original model, I do not think it is appropriate to be regarded as a transfer learning work.
> >
> > Based on the consideration above, I have to keep my rating.

---

> > > ### Author Response · Authors · 2023-11-21
> > > **Response to Reviewer Ka1A**
> > >
> > > We sincerely thank you for the response. We have prepared additional responses for the parts you are not satisfied. If there are still unresolved concerns, please feel free to let us know at any time.
> > >
> > > > 1) why the authors assert that prior works either did not refine prototypes? Indeed, most methods based on prototype learning have to adjust the prototypes for learning better feature representation of each class.
> > >
> > > In many prototype learning works [1], [2], [3], they are applied in few-shot learning scenarios, where prototypes are set based on the features of the support set. In these scenarios, they typically lack an explicit prototype refinement process as the adjusted prototypes can be easily acquired from the updated support set features. In contrast, in transfer learning scenarios, the support set is not commonly used.
> > >
> > > The difference in this setting can affect the quality of the derived prototypes, and our introduced explicit refinement process addresses issues arising from this scenario difference. If one simply substitues support set features with mini-batch features for prototype refinement in transfer learning, it is likely to result in highly unstable prototype update as no features will be included for some classes (e.g., when the number of classes is 555 and the mini-batch size is 64 in NABirds).
> > >
> > >
> > > > 2) since this method does not intend to preserve information from the original model, it is not appropriate to regard it as a transfer learning work.
> > >
> > > Simply preserving information from a pre-trained model is considered suboptimal in the context of transfer learning, as highlighted in various studies [4], [5]. Transfer learning involves adapting the model to perform well on a downstream task, often referred to as "task adaptation" in the literature [6], [7]. Therefore, we believe that adapting the pre-trained model's information to better suit the downstream task is more appropriate than simply preserving it.
> > >
> > > Our work involves leveraging prototypes that incorporate information from the pre-trained model while gradually adapting them. Hence, the statement "does not intend to preserve information" is made because we aim to adapt prototypes and features over time while utilizing the information from the pre-trained model (by aggregating features to initial prototypes).
> > >
> > > References
> > > [1] Snell et al., "Prototypical networks for few-shot learning", NeurIPS 2017
> > > [2] Wang et al., "PANet: few-shot image semantic segmentation with prototype alignment", CVPR 2019.
> > > [3] Cheng et al., "Holistic Prototype Activation for Few-Shot
> > > Segmentation", TPAMI 2023.
> > > [4] Chen et al., "Catastrophic forgetting meets negative transfer: batch spectral shrinkage for safe transfer learning", NeurIPS 2019.
> > > [5] Zhou et al., "DR-Tune: improving fine-tuning of pretrained visual models by distribution regularization with semantic calibration", ICCV 2023.
> > > [6] Zhai et al., "A large-scale study of representation learning with the visual task adaptation benchmark".
> > > [7] Jiang et al., "Transferability in deep learning: A survey".

---

> ### Author Response · Authors · 2023-11-17
> **Response to Reviewer Ka1A (2/3)**
>
> > **Q2: Although the motivation of solving the problem of granularity gap is interesting, I do not quite understand how the proposed mechanism with prototypes helps to eliminate such "gap" under the perspective of transfer learning. In more detail, it seems that this method can also be used to train a randomly initialized model for the fine-grained task, because I do not see a mechanism to "preserve" useful information from the original pre-trained model but only recognize the mechanism to make change to the model.**
>
> A2: We want to clarify that our approach aims not to "preserve" pre-trained information but to focus downstream task-specific details within the pre-trained knowledge. This focus becomes crucial when a significant granularity gap exists, as lower granularity task-specific details might be subtle compared to higher granularity pre-trained information, potentially hindering the transfer of task-specific knowledge.
>
> In the case of a randomly initialized model, the initialized prototypes do not contain any class-discriminative information. Applying ProtoReg to such meaningless prototypes **1) without refinement** would hinder the learning of discriminative information as the downstream model focuses on the meaningless information. However, ProtoReg **2) with prototype refinement** during fine-tuning, the prototypes get updated using discriminative features learned from downstream data. While this ensures that the prototypes contain some discriminative information, it still falls short of fully utilizing the discriminative information within the pre-trained model.
>
> The experiment results below support our argument by demonstrating the performance of applying ProtoReg to a randomly initialized model (with and without prototype refinement). In some cases, ProtoReg brings about a certain degree of performance improvement. However, the significant drop in performance when refinement is not used indicates the importance of prototypes incorporating discriminative information from the pre-trained model.
>
>
> |Methods|Aircraft|Cars|CUB200|NABirds|
> | --- | --- | --- | --- | --- |
> |CE|34.44|43.70|33.03|47.84|
> |ProtoReg (self)|40.23|39.06|39.56|50.55|
> |ProtoReg (self), w/o refine|4.14|9.77|30.24|20.44|
> |ProtoReg (LP)|11.01|0.87|0.50|0.22|
> |ProtoReg (LP), w/o refine|9.18|1.53|2.09|21.64|
>
> [Additional explanation of the mechanism]
> In ProtoReg (LP), the pre-trained encoder is frozen while training classifier weights. The learned classifier weights represent decision boundaries for downstream features obtained from the pre-trained model. These boundaries are drawn based on information distinguishing different classes (these information is with low granularity particularity when the data is fine-grained) in the downstream data. Consequently, explicitly setting prototypes containing discriminative information and aggregating features toward them enables to focus more on low granularity discriminative information. Similarly, ProtoReg (self) employs the use of sample mean to represent discriminative information for each class.
>
> ---
>
> > **Q3: A minor problem: the last sentence of the comments of Figure 1 is confusing. Maybe the word "suffer from" is not correctly used.**
>
> A3: We have made the revision in the manuscript, replacing "suffer from" with the more appropriate phrase "faces challenges in." The changes have been highlighted in red. Thank you for pointing that out.
>
> ---
>
> > **Q4: Considering the 2nd drawback listed above, please explain whether there is a mechanism to discover and preserve useful information from the pre-trained model.**
>
> A4: The question has been answered in the response to Q2. Please kindly refer to that for further details.

---

> ### Author Response · Authors · 2023-11-17
> **Response to Reviewer Ka1A (3/3)**
>
> > **Q5: It is a common scene for real data that each class may have multiple prototypes. How do the authors deal with such a case?**
>
> For ProtoReg (self), instead of using the sample mean when assuming multiple prototypes, we can employ K-means clustering on class-specific features and use the centroids of each cluster as prototypes. However, determining the optimal number of prototypes practically is challenging. We experimentally confirmed that the current approach, using a single prototype per class, is sufficient and achieves the best performance. This is because using multiple prototypes per class may interfere with aggregation.
>
> For ProtoReg (LP), as it relies on the weights of a linear classifier corresponding to the decision boundary, various class-discriminative information are integrated and encoded into a single vector.
>
> ---
>
> > **Q6: Please explain why two different prototype updating strategies are used according to their initialization methods? What is the motivation or consideration of their designs, respectively?**
>
> For our motivation and considerations, we aim to illustrate through the following toy example:
>
> Consider a 2D feature space with two classes, each having two samples, totaling four samples.
>
> Class 1: (-0.1, 0.9), (0.1, 1.1)
>
> Class 2: (1.1, 0.1), (0.9, -0.1)
>
> For ProtoReg (self), prototypes obtained through sample mean are:
>
> Prototype 1: (0, 1.0)
>
> Prototype 2: (1.0, 0)
>
> For ProtoReg (LP), the decision boundary maximizing the margin is y=x, and classifier weight vectors are perpendicular to the decision boundary. Therefore, prototypes are:
>
> Prototype 1: (-$\frac{1}{\sqrt{2}}$, $\frac{1}{\sqrt{2}}$)
>
> Prototype 2: ($\frac{1}{\sqrt{2}}$, -$\frac{1}{\sqrt{2}}$) (prototypes are l2-normalized for simplicity)
>
> In summary, there is a discrepancy between prototypes based on the two initialization methods. Thus, when updating prototypes initialized with linear classifier weights using sample mean, distortion occurs. To prevent this, when initializing with linear classifier weights, prototypes are set to be learnable and updated through backpropagation. On the other hand, when initializing with the sample mean, prototypes are not set to be learnable, and they are updated using the sample mean of the updated features. The reason for not making the sample mean learnable is to utilize features obtained across different iterations comprehensively, ensuring stable training (as a mini-batch may have few or no samples of a specific class).

---

### Official Review · Reviewer_HTua · 2023-11-03

**Soundness:** 3 good
**Presentation:** 3 good
**Contribution:** 2 fair
**Rating:** 3
**Confidence:** 5

**Summary:**

This paper pointed out an issue in fine-grained transfer learning, called the "granularity gap" , i.e., less-discriminative information is also transferred with the discriminative information from a pre-trained model to the downstream model and harmful to the performance of the downstream task. A method called ProtoReg is thus proposed to prioritize fine-grained discriminative information via crafted prototypes, consisting of several main steps including prototype initialization, prototype refinement, and prototype aggregation and separation. Overall, the proposed method sounds technical convincing. Although ‘simple’ is a strong advantage claimed by the authors, the proposed ProtoReg might also come up with some critical drawbacks, e.g., tending to overfit the fine-grained information due to the clustering nature of the proposed method. Besides, the improvement in the performance is not well convincing since the comparing methods are not very up-to-date and basically not focusing on fine-grained problem.

**Strengths:**

The key idea behind the proposed method is reasonable and the method itself is simple to implement.

**Weaknesses:**

· Class-wise prototype generation could lead to overfitting issues.
· Presentations on some key technical contents are not very clear, e.g., why and how exactly the equation (4) is formed.
· Experiments are not well supportive, e.g., the comparing methods are not very up-to-date and basically not focusing on fine-grained problem.

**Questions:**

· How to ensure the generality of the proposed method? To be specific, the discriminative is a relative concept. While the representative prototypes could preserve the main fine-grained information over each class, it does not necessarily preserve marginal fine-grained characteristics of the data, e.g., according to the theory of distributional clustering.
· Computational overhead should be also reported. Although the proposed method is simple to implement, its clustering nature (repeating to compute the 'similarity' between two data points) can make the complexity very high.

---

> ### Author Response · Authors · 2023-11-17
> **Response to Reviewer HTua (1/3)**
>
> > **Q1: Class-wise prototype generation could lead to overfitting issues.**
>
> A1: The essence of transfer learning lies in tuning the model to optimize for the downstream task. Therefore, focusing on discriminative information for a specific downstream task cannot be considered as leading to overfitting. On the contrary, our method is better understood as preventing the downstream model from being biased towards non-discriminative information by focusing on discriminative information.
>
> If prototypes are fixed to specific vectors, the clustering nature might cause features to collapse towards the prototypes, leading to potential overfitting. However, as prototypes are progressively refined to better encompass discriminative information, our method demonstrates robustness against overfitting to fine-grained information or specific prototypes. Numerous experimental results demonstrate that our method is much more robust against overfitting compared to CE:
>
> * Table 1: Significant performance improvements in the limited data regime which is more vulnerable to overfitting, e.g., using only 15% of the training data.
> * Table 5: As training epochs increase, while CE saturates in test accuracy, ProtoReg continues to show a consistent improvement in test accuracy.
> * Figure 13: Training accuracy saturates much more slowly than CE, and validation accuracy consistently improves.
>
> ---
>
> > **Q2: Why and how exactly the equation (4) is formed?**
>
> A2: Thank you for the constructive comments. The following is our response to why and how the equation (4) is formed.
>
> The intuitive approach to obtaining a representative representation for each class is to use the classwise sample mean. Inspired by the widely demonstrated effectiveness of utilizing the sample mean [1], we utilized it not only for initialization but also for refinement.
>
> Moreover, when utilizing downstream data for fine-tuning, features are trained to progressively encode task-discriminative information. As a result, refining prototypes by averaging over samples across progressively shifted feature spaces becomes beneficial for stable training, given the gradual shift towards task-discriminative feature spaces.
>
> Furthermore, the approach is computationally efficient as it directly stores computed features at each iteration, contrasting with the alternative of recomputing the entire set of features at the end of each epoch. Appendix B.7 addresses this point, and Table 10 demonstrates that our approach is robust against overfitting and computationally efficient compared to alternatives.
>
> The detailed process of refinement using Equation (4) is as follows:
>
> * For a downstream task with $N$ classes, $N$ different memory banks (queues) are prepared to hold features of each class.
> * After each iteration, each feature is enqueued into the corresponding memory bank based on its class label.
> * After an epoch of training, each memory bank contains $N_i^{train}$ features, where $N_i^{train}$ is the number of training samples in class $i$.
> * After each epoch, class prototypes are updated based on the sample mean of their corresponding memory bank features.
> * The memory banks are cleared to hold features for the next epoch.
>
> In response to your comment, we have revised the manuscript to enhance the clarity regarding Equation 4, emphasizing it in red. (section 4.3)
>
> Reference
> [1] Snell et al., "Prototypical networks for few-shot learning.", NeurIPS 2017

---

> ### Author Response · Authors · 2023-11-17
> **Response to Reviewer HTua (2/3)**
>
> > **Q3: The comparing methods are not very up-to-date and basically not focusing on fine-grained problem.**
>
> Comparison methods, BSS [1], SN [2], and Co-tuning [3] conducted experiments on Aircraft, Cars, and CUB200, all of which are fine-grained datasets we also experimented with. This ensures a fair comparison. Moreover, although showing substantial improvement on fine-grained transfer by addressing issues due to the granularity gap, our method is not restricted to fine-grained problems. Extending beyond fine-grained problems, we show performance improvements on general datasets such as Caltech101 and CIFAR100 (refer to Table 7 in the Appendix). Additionally, we added comparisons with the two latest works published in 2023, Robust FT ([4], CVPR 2023), and DR-Tune ([5], ICCV 2023), where our methodology substantially outperforms them.
>
> The table below provides a summary of the experimental results, with these results highlighted in red in Table 1 of the revised manuscript.
>
> Aircraft
> |Methods|  15%   | 30% | 50% | 100% |
> | --- | --- | -------- | -------- | -------- |
> |Robust FT|23.91|37.84|49.09|65.08|
> |  DR-Tune   |24.42|40.89|50.65|68.80|
> |ProtoReg (self)|33.66|46.83|59.95|75.25|
> |ProtoReg (LP)|**34.35**|**50.41**|**61.45**|**77.89**|
>
> Cars
> |Methods|  15%   | 30% | 50% | 100% |
> | --- | --- | -------- | -------- | -------- |
> |Robust FT|25.07|52.63|70.10|84.29|
> |  DR-Tune   |26.80|56.04|73.55|84.80|
> |ProtoReg (self)|39.95|65.91|78.36|87.74|
> |ProtoReg (LP)|**42.52**|**69.32**|**81.62**|**89.60**|
>
> CUB200
> |Methods|  15%   | 30% | 50% | 100% |
> | --- | --- | -------- | -------- | -------- |
> |Robust_FT|43.34|57.66|68.35|78.53|
> |  DR_Tune   |44.84|58.49|69.62|79.10|
> |ProtoReg (self)|57.09|67.48|74.53|81.22|
> |ProtoReg (LP)|**59.08**|**69.49**|**76.30**|**82.38**|
>
> NABirds
> |Methods|  15%   | 30% | 50% | 100% |
> | --- | --- | -------- | -------- | -------- |
> |Robust_FT|39.00|55.17|65.46|74.86|
> |  DR_Tune   |41.37|57.07|66.99|76.16|
> |ProtoReg (self)|50.75|63.68|70.84|78.40|
> |ProtoReg (LP)|**52.42**|**64.83**|**72.33**|**79.51**|
>
> References
> [1] Chen et al., "Catastrophic forgetting
> meets negative transfer: Batch spectral shrinkage for safe transfer learning.", NeurIPS 2019
> [2] Kou et al., "Stochastic normalization.", NeurIPS 2020
> [3] You et al., "Co-tuning for transfer learning.", NeurIPS 2020
> [4] Xiao et al., "Masked images are counterfactual samples for robust fine-tuning.", CVPR 2023
> [5] Zhou et al., "DR-Tune: Improving fine-tuning of pretrained visual models by distribution regularization with semantic calibration.", ICCV 2023
>
> ---
>
> > **Q4: How to ensure the generality of the proposed method? To be specific, the discriminative is a relative concept. While the representative prototypes could preserve the main fine-grained information over each class, it does not necessarily preserve marginal fine-grained characteristics of the data, e.g., according to the theory of distributional clustering.**
>
>
> We believe that, for generality, class-common information within fine-grained details is crucial. To ensure class discriminability, it should not merely represent marginal fine-grained characteristics specific to certain data within a class, but rather the main fine-grained information common across the majority of data.
>
> Regarding the generality from the dataset perspective, we would like to clearly emphasize the following: our goal is to prioritize discriminative information. Prioritizing discriminative information can also lead to performance improvement in general classification tasks (Table 7). In fine-grained downstream tasks, as this discriminative information tends to be fine-grained with low granularity (pose challenges for vanilla fine-tuning due to granularity gap), explicitly focusing on them results in more significant performance improvement.

---

> ### Author Response · Authors · 2023-11-17
> **Response to Reviewer HTua (3/3)**
>
> > **Q5: Computational overhead should be also reported. Although the proposed method is simple to implement, its clustering nature (repeating to compute the 'similarity' between two data points) can make the complexity very high.**
>
> In response to you comment, we added a subsection on computational overhead in Appendix B.12 of the revised manuscript, highlighted in red.
> The similarity computation is implemented with a single matrix multiplication, resulting in minimal computational overhead. Specifically, using l2-normalized feature matrix $F \in \mathbb{R}^{B \times d}$ and l2-normalized prototypes matrix $C \in \mathbb{R}^{K \times d}$ (where B is batch size, d is feature dimension, and K is the number of classes), the cosine similarity between each sample and prototype is computed in a single matrix multiplication via $FC^T \in \mathbb{R}^{B \times K}$. The element (i, j) indicates the cosine similarity between the i-th sample and the j-th class prototype.
>
> The table below demonstrates the relative time comparison against CE when experiments are conducted on the same device.
>
> | Method          | Relative Time |
> | --------------- |:-------------:|
> | CE              |       1       |
> | ProtoReg (self) |     1.03      |
> | ProtoReg (LP)   |     1.12      |
>
> ProtoReg (self) has a negligible computational overhead of only 3%, while ProtoReg (LP) has a 12% overhead due to the linear probing process for prototype initialization. In both cases, the overhead is sufficiently low, considering the significant performance improvement they offer.

---

### Meta-Review · Area_Chair_oVJF · 2023-12-05

**Metareview:**

This paper presents an algorithm to fine-tune pre-trained models to fine-grained downstream classification tasks.

After the rebuttal and AC-reviewer discussion stage, the final scores of this paper are 3/5/5/6. Only one reviewer (rating 5) showed up in the discussion, and he/she insisted on the original recommendation, commenting that the paper lacks sufficient technical contributions to get accepted by ICLR. The AC reads the paper and concurs with the majority of reviewers.

**Justification For Why Not Higher Score:**

The average score falls below the acceptance threshold.

**Justification For Why Not Lower Score:**

N/A

---

### Decision · Program_Chairs · 2024-01-16

Reject